# RAS: Retrieval-And-Structuring for Knowledge-Intensive LLM Generation

**Pengcheng Jiang**    **Lang Cao**    **Ruike Zhu**    **Minhao Jiang**    **Yunyi Zhang**
**Jiaming Shen**$^\diamond$    **Jimeng Sun**    **Jiawei Han**

University of Illinois Urbana-Champaign    $^\diamond$Google DeepMind

## Abstract

Large language models (LLMs) have achieved impressive performance on knowledge-intensive tasks, yet they often struggle with multi-step reasoning due to the unstructured nature of retrieved context. While retrieval-augmented generation (RAG) methods provide external information, the lack of explicit organization among retrieved passages limits their effectiveness, leading to brittle reasoning pathways. Recent interpretability studies highlighting the importance of structured intermediate reasoning further align with this perspective. We propose Retrieval-And-Structuring (RAS), a framework that dynamically constructs question-specific knowledge graphs through iterative retrieval and structured knowledge building. RAS interleaves targeted retrieval planning with incremental graph construction, enabling models to assemble and reason over evolving knowledge structures tailored to each query. On seven knowledge-intensive benchmarks, RAS consistently outperforms strong baselines, achieving up to 8.7% and 7.0% gains with proprietary and open-source LLMs, respectively. Our results demonstrate that dynamic, question-specific knowledge structuring offers a robust path to improving reasoning accuracy and robustness in language model generation.

## 1 Introduction

Complex reasoning tasks such as scientific analysis or multi-hop question answering demand both comprehensive knowledge and structured logical thinking (Yang et al., 2018). While large language models (LLMs) have achieved remarkable performance across a wide range of natural language processing tasks (Devlin et al., 2018; Brown et al., 2020), they often struggle with knowledge-intensive reasoning due to the absence of precise, logically organized information (Rae et al., 2021; Ling et al., 2024). This limitation has motivated growing research into augmenting LLMs with structured knowledge to enhance their reasoning capabilities (Wang et al., 2021).

Retrieval-augmented generation (RAG) approaches provide LLMs with additional context from retrieved passages (Guu et al., 2020; Lewis et al., 2020; Izacard & Grave, 2021; He et al., 2024), but often face hallucination challenges (Maynez et al., 2020; Zhang et al., 2023b), where generated content deviates from retrieved information. This issue stems from the unstructured nature of passages, which forces the model to implicitly bridge logical gaps. Briefly, interpretability analyses have suggested that LLMs attempt to chain facts across context internally, and failures in these implicit reasoning chains correlate with hallucinations (Lindsey et al., 2025). These findings reinforce the need for explicitly structured intermediate knowledge to guide reasoning.

Recent efforts have integrated knowledge graphs (KGs) with LLMs (Sun et al., 2019; Yu et al., 2022; He et al., 2024; Edge et al., 2024), providing compact relational representations that support more interpretable reasoning (Hogan et al., 2021; Jiang et al., 2024; Sun et al., 2023). However, existing approaches typically rely on static, corpus-wide graphs. This design introduces two limitations. *First*, global KGs are costly to build and maintain: indexing a corpus like Wikipedia 2018 can require millions of LLM calls and cost tens of thousands to millions of USD (see Appendix G). *Second*, global graphs often blend evidence from many documents, leading to ambiguous or even contradictory relations. For example, a global KG might simultaneously encode that Geoffrey Hinton is linked

to deep learning, to cognitive neuroscience, and to critiques of large models—without clarifying which aspect is relevant to user query's focus. Similarly, biomedical KGs may contain both positive and negative associations between a drug and a disease, reflecting conflicting studies. In contrast, a question-specific KG built from targeted documents resolves these conflicts by grounding relations in a coherent, query-relevant context.

These limitations highlight the need for knowledge graphs that are constructed on demand, tailored to the query, and structured to support reasoning. To this end, we propose **Retrieval-And-Structuring (RAS)**, a framework that dynamically constructs and reasons over question-specific knowledge graphs through iterative retrieval and structured knowledge building. The RAS process unfolds in three steps: (1) a **planning step** that identifies knowledge gaps and generates targeted sub-queries, (2) a **retrieval-and-structuring step** that extracts factual triples from retrieved passages and incrementally builds a question-specific graph, and (3) a **knowledge-augmented answering step** that produces final outputs conditioned on the accumulated structured knowledge.

RAS addresses several limitations of prior methods. Unlike traditional RAG, which performs single-pass retrieval (Guu et al., 2020; Lewis et al., 2020; Izacard & Grave, 2021), RAS iteratively plans and fills knowledge gaps at inference. In contrast to static KG-based approaches (He et al., 2024; Edge et al., 2024), RAS dynamically constructs question-specific graphs tailored to each question, capturing only task-relevant information. This design avoids both the inefficiency of offline indexing and the noise of global graphs, enabling precise and robust reasoning.

Through extensive evaluations across seven benchmarks spanning open-domain QA, closed-set QA, and long-form generation, RAS consistently outperforms strong baselines by 7.0% with open-source LLMs and 8.7% with proprietary models. Our main contributions are:

- We propose RAS, a framework that dynamically builds question-specific knowledge graphs through iterative retrieval and structuring.
- We design a unified graph structure-aware model that jointly plans retrieval and generates answers over evolving knowledge graphs.
- We show consistent gains across seven benchmarks, with up to 8.7% improvement over strong RAG baselines, while maintaining efficiency and scalability.

## 2 RELATED WORK

**Retrieval-Augmented Generation (RAG).** RAG enhances language model performance on knowledge-intensive tasks by incorporating retrieved passages into the model input (Guu et al., 2020; Lewis et al., 2020), improving factual accuracy and grounding. Early approaches retrieved a fixed number of passages once before generation (Shao et al., 2023; Es et al., 2024; Lyu et al., 2024a), while later methods explored adaptive retrieval (Jiang et al., 2023) or retrieval evaluation (Kim et al., 2024b) to improve relevance. Iterative retrieval-generation approaches (Shao et al., 2023; Guan et al., 2024) and targeted subquery strategies (Khattab et al., 2023; Yao et al., 2023; Press et al., 2022; Trivedi et al., 2023) progressively enrich the evidence. Self-RAG (Asai et al., 2023) introduced self-reflective retrieval, and RPG (Lyu et al., 2024b) extracted fine-grained paragraphs. More recent work (Jiang et al., 2025c; Mei et al., 2025) further improves RAG by reinforcement learning over search behaviors. Despite these advances, retrieved context often contains redundancy or misses critical facts. Our work departs from these by converting retrieved content into a structured, evolving graph aligned with the query.

**Graph as Context for LLMs.** Graphs offer explicit, relational structures that help models go beyond flat text by making multi-hop relationships more tractable (Yasunaga et al., 2021; 2022; Yu, 2022; Ju et al., 2022; Zhu et al., 2024; Gutiérrez et al., 2024). GraphToken (Perozzi et al., 2024) shows LLMs can process serialized graphs directly (Liu et al., 2021). G-Retriever (He et al., 2024) leverages global KGs for entity-centric subgraph retrieval, while GraphRAG-style methods (Edge et al., 2024; Jiang et al., 2025b) construct large corpus-level graphs with community summarization. These methods rely on static graphs, which are costly to construct (Appendix G) and often introduce irrelevant noise. By contrast, RAS builds *question-specific knowledge graphs* dynamically, eliminating prohibitive offline costs and providing denser, task-relevant context tailored to each reasoning trajectory. This design aligns with findings that many LLM errors stem from failed implicit reasoning chains (Lindsey et al., 2025), which explicit, query-focused structuring can mitigate.

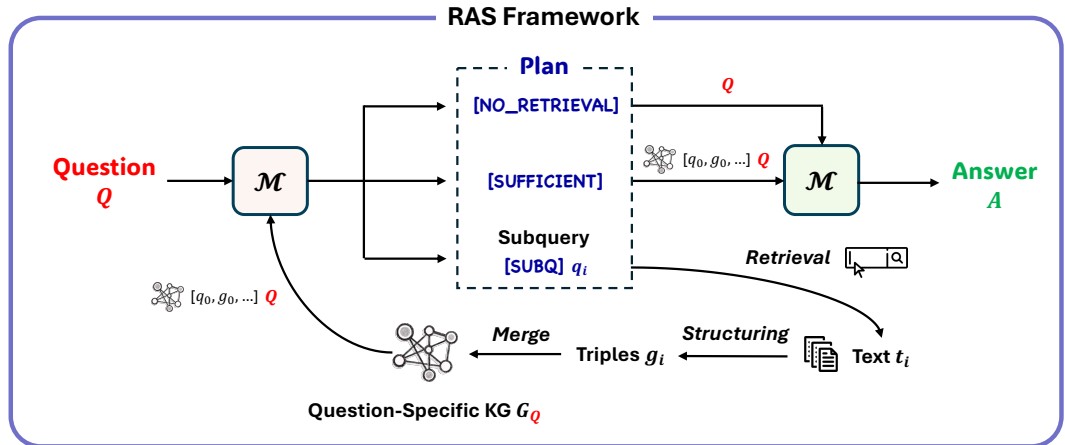

Figure 1: **Overview of the Retrieval-And-Structuring (RAS) framework.** RAS operates through three stages: (1) **Planning (§3.1)**: the model strategically determines retrieval needs and generates focused sub-queries based on the current knowledge state; (2) **Text Retrieval and Structuring (§3.2)**: the system retrieves passages based on sub-queries, extracts factual triples, and merges them into an evolving question-specific knowledge graph that expands iteratively with reasoning needs; and (3) **Answering (§3.3)**: the accumulated structured knowledge is leveraged to generate the final output. We provide a step-by-step running example in Figure 25.

# 3 RETRIEVAL-AND-STRUCTURING (RAS) FRAMEWORK

Effective knowledge-intensive language generation requires not only retrieving relevant information, but also structuring and reasoning over it systematically. We introduce **Retrieval-And-Structuring (RAS)**, a framework that interleaves iterative retrieval planning with dynamic question-specific knowledge graph construction, enabling large language models (LLMs) to reason over progressively organized knowledge tailored to each query. Figure 1 illustrates the overall workflow.

**Key Definitions.** We define the core concepts used in the RAS framework as follows. The *Main question* ($Q$) denotes the original task input. A *subquery* ($q_i$) is a focused retrieval query generated at iteration $i$ to obtain supporting evidence. At the first iteration, $q_0 = Q$. *Retrieved text* ($t_i$) is a set of the top-$k$ documents retrieved by $q_i$. A text-to-triples model $f_{t2t}$ converts retrieved text into *triples* ($g_i$), structured as subject-predicate-object facts. These triples are incrementally accumulated into an evolving *question-specific knowledge graph* ($G_Q$), representing organized evidence related to $Q$. The model $\mathcal{M}$ produces an *plan* ($p_i$) at each step, determining whether to continue retrieval ([SUBQ]), stop retrieval ([SUFFICIENT]), or, initially, answer directly without retrieval ([NO_RETRIEVAL]).

## 3.1 KNOWLEDGE-AWARE PLANNING

The planning step initiates and controls the retrieval-and-structuring process by dynamically assessing the current knowledge state.

**Initial Planning.** Formally, given an input query $Q$, the model $\mathcal{M}$ generates an initial plan $p_0$:

$$p_0 \leftarrow \mathcal{M}(\emptyset; \text{INST}_{\text{Plan}}; \emptyset; Q) \tag{1}$$

where $\text{INST}_{\text{Plan}}$ is the planning instruction (as shown in Figure 15). $p_0$ can take one of two forms:

◇ **[SUBQ]** $q_0 = Q$: If $\mathcal{M}$ assesses that the query cannot be satisfactorily answered with its own knowledge, we start the iteration with the main question $Q$ as the initial subquery, and move to the next stage (§3.2).

◇ **[NO_RETRIEVAL]**: If $\mathcal{M}$ determines that $Q$ can be answered directly without requiring any additional knowledge, the planning process terminates, and the framework proceeds directly to the final Answering stage (§3.3).

**Iterative Planning.** At iteration $i > 0$, given the accumulated knowledge $G_i$ and the subquery-triples history $[q_0, g_0, \ldots, q_i, g_i]$, the model updates the plan:

$$p_{i+1} \leftarrow \mathcal{M}(\text{GNN}(G_i); \texttt{INST}_{\texttt{Plan}}; [q_0, g_0, ..., q_i, g_i]; Q) \qquad (2)$$

where GNN is a graph neural network for encoding and projecting the evolving KG $G_i$; $q_k$ is the subquery at iteration $k$, and $g_k$ is the extracted graph information (a list of triples) from the retrieved context $t_k$.

The output $p_{i+1}$ at each iteration can be either:

⬦ **[SUBQ]** $q_{i+1}$: The model generates a new subquery $q_{i+1}$ to guide the retrieval of additional relevant knowledge. The subquery is designed to fill specific gaps in the current knowledge state with respect to answering $Q$. The framework proceeds to the next stage (§3.2).

⬦ **[SUFFICIENT]**: The accumulated knowledge $G_i$ is deemed sufficient to comprehensively address the main question $Q$. The iterative retrieval process terminates, and the framework proceeds to the Answering stage (§3.3).

Planning serves as a key driver of the RAS framework's iterative retrieval and refinement process. By dynamically assessing the adequacy of the retrieved knowledge and generating targeted sub-queries, it enables the efficient acquisition of query-relevant information.

## 3.2 TEXT RETRIEVAL AND STRUCTURING

Once **[SUBQ]** is detected, we use the subquery $q_i$ to retrieve the text $t_i$ and transform it into structured knowledge $g_i$, which is progressively merged to the question-specific graph $G_Q$.

**Text Retrieval.** We use a text retriever to retrieve the top-$k$ semantically relevant passages $t_i$ from the corpus $C$ for each subquery $q_i$:

$$t_i \leftarrow \text{Retrieval}(q_i, C, k) \qquad (3)$$

We use a standard dense retriever by default but note that RAS is compatible with more advanced information retrieval methods (Chaudhary et al., 2023; Kang et al., 2024; Jiang et al., 2025a).

**Text-to-Triples Conversion.** To extract essential factual information from the retrieved passages $t_i$, we employ a text-to-triples model $f_{t2t}$. This model is trained on the full WikiOfGraph dataset (Kim et al., 2024a), which is a high-quality, LLM-curated text-to-triples corpus. Details of the training process are provided in Appendix D.1. The model generates structured triples in the following format:

$$g_i \leftarrow f_{t2t}(t_i) = [(s_0, r_0, o_0), ..., (s_{|g_i|}, r_{|g_i|}, o_{|g_i|})] \qquad (4)$$

where each triple $(s_j, r_j, o_j)$ represents a subject-predicate-object fact extracted from the text. This structured representation enables efficient downstream reasoning and facilitates integration with external knowledge graphs. Although $f_{t2t}$ is a lightweight LLM capable of fast inference using techniques such as quantization (Dettmers et al., 2022) and optimized inference frameworks like vLLM (Kwon et al., 2023), the text-to-triples conversion can be precomputed offline as well when maximal efficiency is required.

**Iterative Knowledge Enrichment to Question-Specific KG.** The extracted triples $g_i$ are then converted into a graph structure $g_i' = (V_i, E_i)$, where $V_i$ and $E_i$ denote the sets of nodes and edges, respectively. Each node $v \in V_i$ corresponds to a unique subject or object entity in $g_i$, while each edge $e \in E_i$ represents a predicate connecting two entities. To enrich the graph with semantic information, the attributes of nodes and edges are obtained through Sentence-BERT (Reimers, 2019):

$$\text{emb}(v) \leftarrow \text{encode}(v), \forall v \in V_i; \qquad \text{emb}(e) \leftarrow \text{encode}(e), \forall e \in E_i \qquad (5)$$

These semantic embeddings enable the model to capture the nuanced relationships between entities and facilitate reasoning over the KG.

To progressively enrich the question-related knowledge in response to the evolving sub-queries, the structured graph $g_i'$ at each iteration $i$ is merged into an evolving KG $G_Q = (V_Q, E_Q)$ specific to the main question $Q$:

$$G_Q \leftarrow G_Q \cup g_i' \qquad (6)$$

After enriching $G_Q$ with the new knowledge, we plan (§3.1) for the next step. Based on $G_Q$ and the chain of previous subqueries and their associated graph information, the model decides whether to generate another focused subquery for additional retrieval or to proceed with answering (§3.3) if the accumulated knowledge is sufficient.

### 3.3 KNOWLEDGE-AUGMENTED ANSWERING

When answering is triggered, the model $\mathcal{M}$ generates an answer $A$ to the main question $Q$ either conditioned on knowledge graph $G_Q$ and subquery chain $(q_0, g_0), ..., (q_i, g_i)$ when retrieval-and-structuring was processed, or directly when no retrieval is needed.

If no retrieval is needed ($p_0 = [\texttt{NO\_RETRIEVAL}]$), the answer is generated directly:

$$A \leftarrow \mathcal{M}(\emptyset; \texttt{INST}_{\texttt{Ans}}; \emptyset; Q) \tag{7}$$

Otherwise, after iterative knowledge enrichment concludes with $[\texttt{SUFFICIENT}]$ plan or the maximum iteration is reached, the answer is generated using encoded KG ($G_Q$) and subquery chain:

$$A \leftarrow \mathcal{M}(\text{GNN}(G_Q); \texttt{INST}_{\texttt{Ans}}; [q_0, g_0, ..., q_i, g_i]; Q) \tag{8}$$

where $\texttt{INST}_{\texttt{Ans}}$ is the answering instruction (as shown in Figure 16). $\mathcal{M}$ attends to knowledge in $G_Q$ and subquery chain to generate accurate, coherent answers. This structured conditioning enables systematic reasoning grounded in the assembled knowledge.

### 3.4 STRUCTURE-AWARE MULTITASK LEARNING

The RAS framework is trained through a multitask setup that unifies knowledge-aware planning and knowledge-augmented answering under a standard next-token prediction objective.

Each training instance corresponds to either a planning task or an answering task, selected randomly:

- ◇ **Planning**: Input: current encoded question-specific KG $G_Q$, planning instruction $\texttt{INST}_{\texttt{Plan}}$ (shown in Figure 15), subquery-triples history ($[q_0, g_0, ..., q_i, g_i]$), main question $Q$. Output: next plan $p_{i+1}$.
- ◇ **Answering**: Input: final encoded question-specific KG $G_Q$, answering instruction $\texttt{INST}_{\texttt{Ans}}$ (shown in Figure 16), subquery-triples history ($[q_0, g_0, ..., q_i, g_i]$), main question $Q$. Output: final answer $A$.

The model $\mathcal{M}$ used in RAS is based on *Graph LLM*, an architecture adapted from prior work (Perozzi et al., 2024; He et al., 2024). *Note: This multitask training setup is applied to open-source setting, while we test RAS under closed-source setting (see Appendix F.1) in Section 4 as well.*

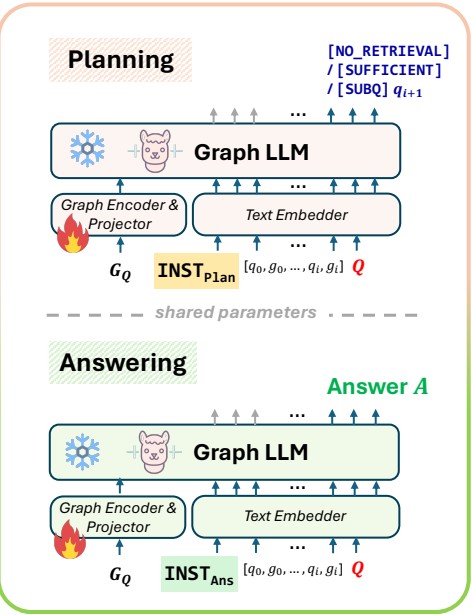

Figure 2: **Structure-Aware Multitask Learning for the Graph LLM in RAS.** A single LLM is trained with both planning and answering tasks in a parameter-efficient way (fine-tuning the graph components with LoRA (Hu et al., 2022)).

## 4 EXPERIMENTS

### 4.1 SETTINGS

**Training Data & Setting.** We develop *HotpotQA-SUBQ*, a dataset constructed based on HotpotQA (Yang et al., 2018), to train our model's action planning and answering capabilities. Our dataset creation begins with document filtering: using Claude-3.5-Sonnet (Anthropic, 2024), we identify and retain only the supporting documents necessary for answering the main question, removing irrelevant content. For each supporting document $d_j$, we then iteratively generate a subquery $q_j$, considering the main question, previous subqueries, and supporting documents. During iteration $j$, when more supporting documents remain, we create training samples with input $\{q_0, g_0, ..., q_j, g_j, Q\}$ and output label "$[\texttt{SUBQ}]$ $q_{j+1}$", where $g_k$ represents triples extracted from document $d_k$, and $Q$ is the main question. For the final supporting document, we label the sample as "$[\texttt{SUFFICIENT}]$". To identify queries that can be answered directly, we test our base LLM (LLaMA-2-7B) on HotpotQA's main

queries without context. For correctly answered queries, we create training samples with the main question $Q$ as input and "`[NO_RETRIEVAL]`" as the output label. Additionally, to ensure fair comparison with existing approaches, we incorporate the subset of Arc-Easy (2,147 samples) and ASQA (3,897 samples) from Self-RAG's training data, resulting in 208k training samples in total. We place detailed training data processing, dataset statistics, and data samples in Appendix C. For efficient inference, we train a text-to-triples model $f_{t2t}$ on the WikiOFGraph dataset (Kim et al., 2024a) using LLaMA-3.2-3B-Instruct as the base model (see Appendix D), and deploy it using vLLM for optimized runtime performance. We present our hyperparameter study of each component in Appendix J. **Knowledge Sources.** We employ faiss (Douze et al., 2024) for efficient vector searching over the dense index. Following the Self-RAG (Asai et al., 2023), we utilize the Wikipedia 2018 (Izacard et al., 2023) by default, while specifically using the Wikipedia 2020 for PopQA to access more recent information. To optimize retrieval efficiency, we partition the index into five segments.

**Test Datasets & Metrics & Compared Baselines.** We conduct comprehensive evaluations on diverse knowledge-intensive tasks following previous studies (Asai et al., 2023; Lyu et al., 2024b). The evaluation encompasses three categories of datasets: (1) open-domain short-form generation datasets: TriviaQA (Joshi et al., 2017), PopQA (Mallen et al., 2022), and 2WikiMultihopQA (Ho et al., 2020); (2) closed-set task datasets: PubHealth (Zhang et al., 2023a) and ARC-Challenge (Clark et al., 2018); and (3) long-form generation datasets: ALCE-ASQA (Gao et al., 2023; Stelmakh et al., 2022) and ELI5 (Fan et al., 2019). For evaluation metrics, we maintain consistency with prior work (Asai et al., 2023; Mallen et al., 2022; Lyu et al., 2024b), employing "golden match" accuracy for PopQA and TriviaQA, token-level F1 score for 2WikiMultihopQA, accuracy for PubHealth and ARC-Challenge, and ROUGE-LSum alongside MAUVE score (Pillutla et al., 2021) for ASQA and ELI5. Our comparative analysis includes three baseline categories: models without retrieval augmentation, incorporating Claude 3.5 Sonnet as a state-of-the-art closed-source baseline; models with single retrieval over top-5 documents, including Claude 3.5 Sonnet, and SuRe (Kim et al., 2024b), a leading retrieve-and-summarize method; and models with self-reflective retrieval, including leading approaches Self-RAG (Asai et al., 2023), RPG (Lyu et al., 2024b) . We place more details of datasets and metrics in Appendices E.1 and E.2, respectively.

**Inference Setting.** We evaluate RAS using both our trained open-source models ($RAS_{7B/8B}$, based on LLaMA-2-7B/LLaMA-3-8B and a Graph Transformer encoder (Shi et al., 2020)) and the closed-source Claude-3.5-Sonnet & Claude-4.5-Sonnet (Anthropic, 2025) under varied inference settings. For open-source models ($RAS_{7B/8B}$), we follow prior work (Asai et al., 2023; Lyu et al., 2024b) and adopt zero-shot inference across all datasets. For the closed-source model ($RAS_{Sonnet-3.5}$, $RAS_{Sonnet-4.5}$), we apply few-shot inference with two exemplars for ASQA and ELI5, while using zero-shot inference for all other datasets. More details are provided in Appendices F.1 and F.2.

For PopQA and TriviaQA evaluation, we follow established settings (Asai et al., 2023; Luo et al., 2023a), incorporating top-five web search engine results as initial retrieved context $t_0$. For ASQA and ELI5, we maintain methodological consistency with prior work (Lyu et al., 2024b; Asai et al., 2023; Gao et al., 2023), utilizing their *predetermined five-document context*. Under these conditions, we omit plan generation and text retrieval phases, implementing static inference (Asai et al., 2023) with five fixed iterations. For remaining datasets, we establish a maximum iteration count of five. Following previous studies, we employ Contriever-MS MARCO (Izacard et al., 2021) as the primary dense retriever, with BM25 (Robertson et al., 2009) serving as the retrieval mechanism for 2WikiMultihopQA. Across all retrieval processes, we maintain a consistent top-$k$ document selection of five. We present more details of inference settings in Appendix F.

## 4.2 RESULTS

**Main Result.** Our performance evaluation, presented in Table 1, demonstrates that the LLaMA-2-7B/LLaMA-3-8B model fine-tuned with RAS outperforms existing SFT-based open-source solutions, including Self-RAG (Asai et al., 2023) and RPG (Lyu et al., 2024b). Notably, compared to the earlier SOTA models, $RAS_{7B}$ shows a 9.7% improvement in short-form question-answering and a 7.9% gain in long-form generation tasks. Additionally, when applied to Claude-3.5-Sonnet, RAS consistently achieves superior results compared to single retrieval RAG approaches, including retrieve-and-summarize approach SuRe (Kim et al., 2024b). We find that sometimes (e.g., on TriviaQA and PubHealth) single-hop retrieval could not boost LLM's performance, and even makes it worse, which demonstrates the necessity of on-demand retrieval, aligning with previous findings.

| | | Short-form | | | Closed-set | | Long-form Generation | | | |
|---|---|---|---|---|---|---|---|---|---|---|
| | Model/Method | TQA (acc) | 2WQA (F1) | PopQA (acc) | Pub (acc) | ARC (acc) | ASQA (rouge) | ASQA (mauve) | ELI5 (rouge) | ELI5 (mauve) |
| **Closed-source** | **w/o Retrieval** | | | | | | | | | |
| | ChatGPT | 74.3 | 24.8 | 29.3 | 70.1 | 75.3 | 36.2 | 68.8 | 22.8 | 32.6 |
| | Sonnet-3.5 | 78.4 | 40.0 | 30.2 | 83.7 | 88.5 | 37.0 | 39.1 | 21.8 | 26.5 |
| | **w/ Single Retrieval (#docs=5)** | | | | | | | | | |
| | Sonnet-3.5$_{\#docs=1}$ | 69.1 | 41.9 | 51.5 | 49.1 | 88.6 | n/a | n/a | n/a | n/a |
| | Sonnet-3.5$_{\#docs=5}$ | 72.5 | 53.7 | 57.3 | 53.9 | 87.1 | 38.8 | 61.6 | 20.2 | 32.3 |
| | SuRe$_{GPT-4o}$ (Kim et al., 2024b) | 72.3 | 38.1 | 53.6 | 57.2 | 79.6 | 36.0 | 74.2 | 19.2 | 51.6 |
| | SuRe$_{Sonnet-3.5}$ | 76.8 | 37.6 | 41.2 | 62.8 | 91.6 | 30.2 | 69.9 | 15.4 | 27.2 |
| | **w/ Self-Reflective Retrieval** | | | | | | | | | |
| | ReAct$_{Sonnet-3.5}$ (Yao et al., 2023) | 73.4 | 53.7 | 55.0 | 62.2 | 89.2 | 38.8 | 61.6 | 20.2 | 32.3 |
| | IRCoT$_{Sonnet-3.5}$ (Trivedi et al., 2023) | 74.7 | 54.9 | 53.2 | 59.4 | 92.0 | 38.8 | 61.6 | 20.2 | 32.3 |
| | **Retrieval-And-Structuring (ours)** | | | | | | | | | |
| | RAS$_{Sonnet-3.5}$ | 77.6 | 57.7 | 62.3 | 71.3 | 93.9 | 39.1 | 70.5 | 23.3 | 37.7 |
| **Open-source** | **w/o Retrieval** | | | | | | | | | |
| | Llama2$_{7B}$ | 30.5 | 18.9 | 14.7 | 34.2 | 21.8 | 15.3 | 19.0 | 18.3 | 32.4 |
| | Llama2$_{13B}$ | 38.5 | 20.2 | 14.7 | 29.4 | 29.4 | 12.4 | 16.0 | 18.2 | 41.4 |
| | Llama3$_{8B}$ | 56.1 | 21.2 | 26.7 | 33.2 | 42.2 | 17.6 | 25.0 | 18.2 | 39.7 |
| | **w/ Single Retrieval (#docs=5)** | | | | | | | | | |
| | Llama2$_{7B}$ | 42.5 | 21.0 | 38.2 | 30.0 | 48.0 | 22.1 | 32.0 | 18.6 | 35.3 |
| | Llama2$_{13B}$ | 47.0 | 31.2 | 45.7 | 30.2 | 26.0 | 20.5 | 24.7 | 18.6 | 42.3 |
| | Llama3$_{8B}$ | 60.4 | 33.4 | 48.6 | 36.5 | 40.1 | 23.9 | 52.1 | 18.8 | 40.7 |
| | SuRe$_{7B}$ (Kim et al., 2024b) | 51.2 | 20.6 | 39.0 | 36.2 | 52.7 | 35.8 | 76.2 | 16.1 | 26.6 |
| | **w/ Self-Reflective Retrieval** | | | | | | | | | |
| | Self-RAG$_{7B}$ (Asai et al., 2023) | 66.4 | 25.1 | 54.9 | 72.4 | 67.3 | 35.7 | 74.3 | 17.9 | 35.6 |
| | Self-RAG$_{13B}$ | 69.3 | 26.9 | 55.8 | 74.5 | 73.1 | 37.0 | 71.6 | 18.7 | 38.5 |
| | RPG$_{7B}$ (Lyu et al., 2024b) | 65.1 | 33.6 | 56.0 | 73.4 | 65.4 | 37.6 | 84.4 | 19.1 | 46.4 |
| | ReAct$_{7B}$ (Yao et al., 2023) | 64.0 | 25.0 | 42.7 | 52.4 | 59.0 | 22.1 | 32.0 | 18.6 | 35.3 |
| | IRCoT$_{7B}$ (Trivedi et al., 2023) | 61.5 | 27.6 | 44.3 | 59.6 | 61.6 | 22.1 | 32.0 | 18.6 | 35.3 |
| | **Retrieval-And-Structuring (ours)** | | | | | | | | | |
| | RAS$_{7B}$ | 72.7 | 42.1 | 58.3 | 74.7 | 68.5 | 37.2 | 95.2 | 19.7 | 47.8 |
| | RAS$_{8B}$ | 73.8 | 44.2 | 57.7 | 77.6 | 71.4 | 37.6 | 96.2 | 20.1 | 54.4 |

Table 1: **Performance Comparison.** We highlight the top-2 closed-source models and top-2 open-source 7B models, with the best model in each category underlined for each dataset. RAS is designed for general open-domain question answering rather than Knowledge Graph Question Answering (KGQA) tasks (Perevalov et al., 2022); we therefore exclude comparisons with KGQA-specific methods (Sun et al., 2023; Luo et al., 2023b; Ma et al., 2024). Graph encoding and projection are omitted for RAS$_{Sonnet-3.5}$.

Although RAS integrates planning and answering in a unified framework, these components can be decoupled for greater flexibility. Our "role-swapping" study in Figure 3 demonstrates that performance is primarily constrained by answering capability rather than planning. When Sonnet-3.5 handles planning while RAS$_{7B}$ performs answering, the system achieves 62.4% accuracy on ARC-C, compared to 93.9% when Sonnet-3.5 performs both tasks. Despite having 60× fewer parameters, RAS$_{7B}$ achieves planning performance comparable to (and sometimes exceeding) Sonnet-3.5.

We analyze the impact of each component of RAS in Table 2 and provide detailed discussion below.

**Effect of Iterative Planning and Retrieval.** Comparing the base model with "No Planning" variant shows that iterative planning provides consistent improvements across all metrics (e.g., +8.8% on TQA, +9.0% on 2WQA). This demonstrates the importance of dynamically determining retrieval needs and generating focused sub-queries. Without planning, the model relies on single-pass retrieval, which may miss crucial information needed for complex reasoning. Also, when turning off retrieval, the performance degradation is more obvious, due to the knowledge-intensive nature of those datasets.

**Effect of Graph Construction and Encoding.** The impact of structured knowledge representation is evident from two ablations. First, "No Text-to-Triple" degrades performance significantly on all metrics (e.g., -9.0% on 2WQA, -22.2% MAUVE on ASQA), showing the value of essential information extraction via converting retrieved text into structured triples. Second, removing the graph encoder ("No GraphToken") during training or inference consistently hurts performance across datasets, with particularly large drops on PubHealth (-11.2% and -10.0% respectively). This suggests that the graph structure encoding helps the model better leverage the knowledge relationships.

|  | TQA | 2WQA | Pub | ASQA | |
|---|---|---|---|---|---|
|  | (acc) | (F1) | (acc) | (rg) | (mv) |
| RAS$_{7B}$ | **72.7** | **42.1** | **74.7** | **37.2** | **95.2** |
| *Training Phase* | | | | | |
| w/o GraphEncode | 70.2 | 38.4 | 66.4 | 33.1 | 85.0 |
| w/o LoRA | 71.5 | 37.8 | 54.8 | 32.8 | 84.8 |
| w/o Text-to-Triple | 70.4 | 38.2 | 71.4 | 36.2 | 73.8 |
| w/o Multi-Task | 68.6 | 39.2 | 65.5 | 36.7 | 88.9 |
| *Inference Phase* | | | | | |
| w/o Retrieval | 56.9 | 27.4 | 69.0 | 31.3 | 70.6 |
| w/o GraphEncode | 68.8 | 38.7 | 67.3 | 36.5 | 93.6 |
| w/o Planning | 66.7 | 37.8 | 71.5 | 37.2 | 95.2 |

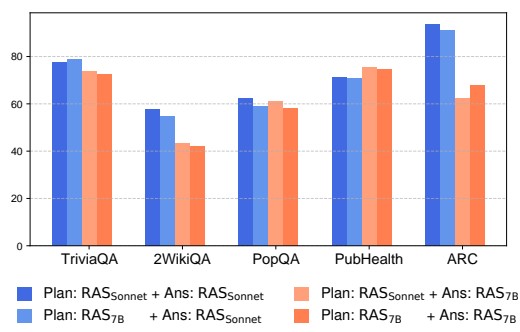

Figure 3: **Role-Swapping Study.** We alternate Sonnet-3.5 and RAS$_{7B}$ on the planning and answering tasks, and evaluate the overall performance.

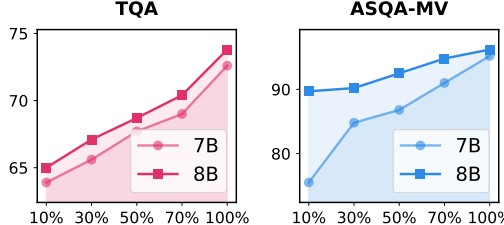

Table 2: **Ablations in Training and Inference (with RAS$_{7B}$).** Training: "No GraphEncode" removes the graph encoder during training, using only LoRA-based LLM fine-tuning. "No LoRA" uses graph token optimization without low-rank adaptation. "No Text-to-Triple" keeps the original retrieved texts instead of converting them into triples. "No Multi-Task" trains two models separately handling planning and answering. **Inference**: "No Retrieval" tests direct query answering without any context. "No GraphEncode" removes graph encoding and projection during inference, using only textual context. "No Planning" removes the planning module and runs single-pass retrieval-structuring-answering pipeline.

Figure 4: **Impact of Graph Information Abundance.** For each sample, we randomly shuffle its associated triples five times and take different ratios (10%–100%) of the shuffled data. The performance scores are averaged across these five shuffling runs.

**Effect of LoRA and Multi-task Learning.** Our experiments reveal that parameter-efficient training strategies significantly impact model performance. Using only graph token optimization without LoRA leads to substantial degradation (-11.8% on average). A similar observation can be made for "No Multi-Task" (e.g., 12.3% accuracy degradation on PubHealth), indicating the significance of jointly training the model on both action planning and answer generation tasks rather than optimizing for each task separately, supporting findings from prior work (Lyu et al., 2024b). The complementary effects suggest that while graph-based knowledge representation is valuable, it needs to be combined with careful parameter tuning and multi-task learning to achieve optimal performance.

**Impact of Graph Information Abundance.** Figure 4 shows that increasing the amount of structured graph information consistently improves RAS's performance across tasks. For TQA, both RAS$_{7B}$ and RAS$_{8B}$ exhibit approximately linear gains as more triples are retained, with no clear saturation even at 100% information. For ASQA-MV, RAS$_{7B}$ benefits from a sharp improvement between 10% and 30% graph information, followed by steady increases, while RAS$_{8B}$ maintains a smoother and more stable growth pattern throughout. These results confirm that RAS effectively leverages structured knowledge at all levels of availability, and that more complete knowledge graphs consistently translate to stronger reasoning and generation quality. Additionally, the larger 8B model consistently outperforms the 7B variant under all conditions, suggesting that scaling the model size further enhances RAS's ability to utilize structured knowledge. Interestingly, even partial graphs containing only 30%–50% of triples already deliver substantial gains over low-information baselines, highlighting RAS's robustness to incomplete or partially retrieved knowledge.

**Impact of Training Data Volume.** Figure 5 demonstrates how training dataset size influences model performance across different tasks. Considering the computational efficiency, we sampled 2,000 instances each from TQA and 2WQA for evaluation, while maintaining the original sizes of other datasets. The results indicate that model performance generally improves with increased training data volume. Notably, our model achieves competitive performance even with limited training data - using only 5% (10K instances) of our full dataset, it surpasses previous state-of-the-art models on TQA, 2WQA, and ELI5. These results suggest both the robustness of our ar-

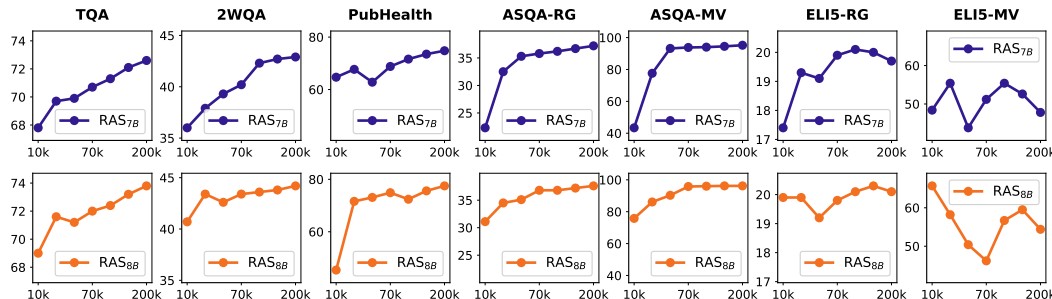

Figure 5: **Impact of Training Data Volume on Model Performance.** Results for RAS$_{7B}$ (top) and RAS$_{8B}$ (bottom) illustrate how performance scales with increasing training data.

chitectural design and the effectiveness of our data curation methodology. However, we observed an exception with the ELI5 dataset, where performance were inconsistent. This irregularity can be attributed to the inclusion of ASQA training data in our training set, following established setting from previous research (Asai et al., 2023; Lyu et al., 2024b). Among our test datasets, ASQA and ELI5 are unique in requiring long-form response generation. The periodic decline in ELI5 performance suggests that the model's response generation began to align more closely with ASQA's training data distribution, potentially at the expense of ELI5's distinct characteristics.

**Impact of Triple Extractor Selection** We examine how triple extraction quality affects RAS performance using three models: Flan-T5-Large, LLaMA-3.2-3B, and Claude-3.5-Sonnet. As shown in Table 3, higher-quality extraction consistently improves results across datasets, with Claude-3.5-Sonnet achieving the best accuracy but at significantly lower efficiency (1.5e-

| Triple Extractor $f_{t2t}$ | Performance | | | Efficiency |
| | TQA | 2WQA | PopQA | (tokens/s) |
| --- | --- | --- | --- | --- |
| Flan-T5-Large | 70.3 | 40.7 | 56.7 | 166.0 |
| LLaMA-3.2-3B | 72.7 | 42.1 | 58.3 | **4,885.3** |
| Claude-3.5-Sonnet | **73.8** | **44.5** | **60.1** | 68.2 |

Table 3: **Impact of Triple Extractor Selection.** Higher-quality text-to-triples models lead to better answer generation in RAS. Claude-3.5-Sonnet achieves the best performance on TriviaQA, 2WikiMultihopQA, and PopQA. LLaMA-3.2-3B (deployed w/ vLLM) is used in our experiments, selected for its strong balance between accuracy and efficiency.

2e-4 sec/token vs 2.0e-4 for LLaMA-3.2-3B). These results demonstrate the accuracy-efficiency tradeoff in triple extraction, with LLaMA-3.2-3B providing the optimal balance for our experiments.

## 5    CONCLUSION

We presented RAS, a framework that dynamically constructs question-specific knowledge graphs through iterative retrieval and structured reasoning. Experiments across seven benchmarks show consistent improvements of up to 6.4% and 7.0% with open-source and proprietary LLMs, respectively. The modular architecture enables transparent reasoning and seamless integration with external knowledge sources. Limitations and future work are discussed in Appendix A. The statement of the LLM usage is placed in Appendix B. Broader Impacts, safeguards, and used assets are discussed in Appendix K. Our data and code can be found at: https://github.com/pat-jj/RAS.

## ETHICS STATEMENT

This work does not involve human subjects, personal data, or sensitive user information. Our research focuses on algorithmic improvements for knowledge-intensive language model reasoning. We are mindful of potential ethical concerns, such as risks of misinformation or biased outputs when deploying retrieval-augmented LLMs. To mitigate these risks, we provide transparent descriptions of our methods, ablation studies on failure cases, and open-sourcing of code and models to facilitate community scrutiny. We also discuss broader impacts and safeguards in Appendix K.

ACKNOWLEDGMENT

Research was supported in part by a research gift from Google Inc.; the AI Institute for Molecular Discovery, Synthetic Strategy, and Manufacturing: Molecule Maker Lab Institute, funded by U.S. National Science Foundation under Awards No. 2019897 and 2505932; NSF IIS 25-37827; and the Institute for Geospatial Understanding through an Integrative Discovery Environment (I-GUIDE) by NSF under Award No. 2118329. The research has used the Delta/DeltaAI advanced computing and data resource, supported in part by the University of Illinois Urbana-Champaign and through allocation #250851 from the Advanced Cyberinfrastructure Coordination Ecosystem: Services & Support (ACCESS) program, which is supported by National Science Foundation grants OAC #2320345, #2138259, #2138286, #2138307, #2137603, and #2138296.

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

**Contents of Appendix**

## A LIMITATIONS & FUTURE WORK

While RAS demonstrates strong performance, several key limitations warrant consideration for future improvements:

- **Knowledge Extraction Quality:** The effectiveness of both planning and answer generation depends on the quality of text-to-triple extraction. Despite using an end-to-end text-to-triples model for triple extraction, implementing a more sophisticated approach – such as a dedicated Named Entity Recognition and Relation Extraction (NER-RE) pipeline – could potentially enhance RAS's reasoning capabilities and knowledge representation.

- **Graph Evolution Strategy:** The current approach to evolving knowledge graphs could be enhanced by incorporating more sophisticated graph pruning and merging strategies. Future work could explore dynamic graph summarization techniques to maintain the most relevant information while preventing excessive graph growth during iterations.

- **Cross-Domain Adaptability:** While RAS performs well on the evaluated tasks, its effectiveness across highly specialized domains or multilingual settings remains to be investigated. Future research could focus on developing domain-adaptation techniques and multilingual knowledge structuring approaches.

- **Interactive Refinement:** The current implementation lacks mechanisms for interactive refinement of the constructed knowledge graphs. Future versions could incorporate human feedback loops to improve the quality of knowledge structuring and reasoning paths.

These limitations suggest several promising directions for future research, including improved knowledge extraction techniques and enhanced graph evolution strategies. A particularly promising direction is the integration of external knowledge graphs into RAS's iterative loop, as its graph-based architecture naturally accommodates the incorporation of established KGs like Wikidata or domain-specific knowledge bases. This could significantly enhance the system's reasoning capabilities by combining dynamically constructed knowledge with curated structured information.

## B THE USE OF LARGE LANGUAGE MODELS

In this work, large language models (LLMs) were used *solely for drafting and polishing the manuscript*. Their role was limited to grammar correction, phrasing refinement, and enhancing readability. All research ideas, methodological innovations, theoretical analyses, and experimental work were entirely conceived, designed, and carried out by the authors. The authors take full responsibility for the scientific content of the paper.

## C    HOTPOTQA-SUBQ DATASET

The source of our dataset is HotpotQA (Yang et al., 2018).[1] Our goal is to create a dataset that can let the model learn the capabilities of subquery generation and answering with intensive knowledge. We name the dataset as HotpotQA-SUBQ.

The construction of HotpotQA-SUBQ includes three steps: (1) filter out irrelevant/unnecessary "supporting docs" (§C.1), (2) generate one subquery for each supporting document (§C.2), and (3) sample labeling (§C.3).

### C.1    SUPPORTING DOCUMENT FILTERING

To identify and filter out irrelevant supporting documents from HotpotQA, we employ an instruction-based approach using Claude 3.5 Sonnet, as shown in Figure 6.

---

Identify which documents are HELPFUL to answer the question. Output only the document numbers separated by commas.

Examples:

Example 1 (Some documents are not helpful):
Question: What nationality was James Henry Miller's wife?
Supporting docs:
1. Margaret "Peggy" Seeger (born June 17, 1935) is an American folksinger. She is also well known in Britain, where she has lived for more than 30 years, and was married to the singer and songwriter Ewan MacColl until his death in 1989.
2. Seeger's father was Charles Seeger (1886–1979), an important folklorist and musicologist; her mother was Seeger's second wife, Ruth Porter Crawford.
3. James Henry Miller, better known by his stage name Ewan MacColl, was an English folk singer and songwriter.
Output: 1,3
Explanation: Only docs 1 and 3 are helpful – doc 1 shows Peggy Seeger (who is American) was married to Ewan MacColl, and doc 3 confirms Ewan MacColl is James Henry Miller. Doc 2 about Seeger's parents is not helpful.

Example 2 (All documents are helpful):
Question: The Oberoi family is part of a hotel company that has a head office in what city?
Supporting docs:
1. The Oberoi family is an Indian family that is famous for its involvement in hotels, namely through The Oberoi Group.
2. The Oberoi Group is a hotel company with its head office in Delhi.
Output: 1,2
Explanation: Both docs are helpful – doc 1 links the Oberoi family to The Oberoi Group, and doc 2 provides the head office location.

Question: **[question]**
Supporting docs:
**[enumerated_documents]**

Output only the helpful document numbers separated by commas:

---

Figure 6: **Prompt used for filtering supporting documents in HotpotQA.** The prompt includes examples to demonstrate the difference between helpful and irrelevant documents. The input parts to the prompt are highlighted.

The filtering prompt is designed with clear examples that illustrate the criteria for document relevance. For each HotpotQA sample, we enumerate all supporting documents and use Claude to identify only

---

[1]We use the version `hotpot_train_v1.1` from https://github.com/hotpotqa/hotpot.

those that contribute directly to answering the question. After filtering out irrelevant documents, we filter the question with no supporting documents. This filtering step reduces noise in the training data and helps focus the model on truly relevant information during sub-query generation.

## C.2   SUB-QUERY GENERATION

For sub-query generation, we use the template as follows to generate one subquery per (filtered) document:

---

Given this main question and a supporting document, generate a simple sub–query (a question) that will help retrieve information from the document to answer the main question.

Main Question: **[main_question]**

Current Document ([topic]):
**[document_content]**

[If previous queries exist:]
Previously generated sub–queries:
 - **[sub_query_1]**
 - **[sub_query_2]**
 ...

Write ONE clear and specific question that:
1. Can be answered using ONLY this document
2. Helps retrieve information needed for the main question
3. Is direct and focused on key information from this document

Write only the question, without any explanations or formatting.

---

Figure 7: **Prompt used for subquery generation from HotpotQA.** The input parts to the prompt are highlighted.

The prompt template enforces document-specificity, goal-orientation, and conciseness in sub-query generation. For iterative querying, we maintain a list of previously generated sub-queries to avoid redundancy and encourage progressive information gathering.

## C.3   SAMPLE LABELING

The sample labeling process transforms the filtered and subquery-augmented HotpotQA examples into training instances for both the Planning and Answering components. We formalize this process in Algorithm 1.

The algorithm takes as input the filtered HotpotQA dataset $\mathcal{D}$, base language model $\mathcal{M}$, and text-to-triple conversion model $f_{t2t}$. For each example, we first attempt direct answer generation using the base LLM (Line 3). If successful, we create [NO_RETRIEVAL] training instances for both the planning and answering (Lines 4-6).

For examples requiring retrieval, we process each supporting document sequentially:

1. Convert document text to structured triples using $f_{t2t}$ (Line 12)
2. For intermediate documents ($i < n$):
    - Construct input by concatenating previous subquery-graph pairs

---

**Algorithm 1** HotpotQA-SUBQ Sample Labeling

---

**Require:** $\mathcal{D}$: Filtered HotpotQA dataset
**Require:** $\mathcal{M}$: Base LLM (LLaMA-2-7B)
**Require:** $f_{t2t}$: Text-to-triple conversion model
**Ensure:** $\mathcal{T}_{plan}$: Training data for Planning
**Ensure:** $\mathcal{T}_{ans}$: Training data for Answering
 1: Initialize $\mathcal{T}_{plan}, \mathcal{T}_{ans} \leftarrow \{\}, \{\}$
 2: **for all** $(Q, \{d_0, ..., d_n\}, A) \in \mathcal{D}$ **do**
 3:     $\hat{A} \leftarrow \mathcal{M}(Q)$                  $\hookleftarrow$ Direct answer attempt
 4:     **if** $\hat{A} = A$ **then**
 5:        $\mathcal{T}_{plan} \leftarrow \mathcal{T}_{plan} \cup \{(Q, \text{[NO\_RETRIEVAL]})\}$
 6:        $\mathcal{T}_{ans} \leftarrow \mathcal{T}_{ans} \cup \{(Q, A)\}$
 7:        continue
 8:     **end if**
 9:     $subq_0, ..., subq_n \leftarrow \text{GenerateSubqueries}(Q, \{d_0, ..., d_n\})$
10:     $G_0, ..., G_n \leftarrow \emptyset$             $\hookleftarrow$ Initialize graph contexts
11:     **for** $i \leftarrow 0$ **to** $n$ **do**
12:        $g_i \leftarrow f_{t2t}(d_i)$             $\hookleftarrow$ Convert text to triples
13:        **if** $i < n$ **then**
14:           $input \leftarrow \text{FormatInput}(\{(subq_j, g_j)\}_{j=0}^{i}, Q)$
15:           $\mathcal{T}_{plan} \leftarrow \mathcal{T}_{plan} \cup \{(input, \text{[SUBQ]} \; subq_{i+1})\}$
16:        **else**
17:           $input \leftarrow \text{FormatInput}(\{(subq_j, g_j)\}_{j=0}^{n}, Q)$
18:           $\mathcal{T}_{plan} \leftarrow \mathcal{T}_{plan} \cup \{(input, \text{[SUFFICIENT]})\}$
19:           $\mathcal{T}_{ans} \leftarrow \mathcal{T}_{ans} \cup \{(input, A)\}$
20:        **end if**
21:        $G_i \leftarrow G_{i-1} \cup g_i$          $\hookleftarrow$ Accumulate graph context
22:     **end for**
23: **end for**
24: **return** $\mathcal{T}_{plan}, \mathcal{T}_{ans}$

---

- Label with [SUBQ] and next subquery
3. For final document ($i = n$):
  - Include all accumulated context
  - Label planning sample as [SUFFICIENT]
  - Create answering training instance with ground truth answer

The input formatting function (Lines 14 and 17) follows the template:

```
[SUBQ] q0
Retrieved Graph Information: g0
[SUBQ] q1
Retrieved Graph Information: g1
...
Question: Q
```

This process generates two datasets:

- $\mathcal{T}_{plan}$: Trains the planning ability to determine retrieval needs and generate targeted sub-queries
- $\mathcal{T}_{ans}$: Trains the answering ability to synthesize final responses from accumulated graph context

Table 4 illustrates the distribution of our generated datasets. The planning dataset demonstrates a balanced distribution across three label types: 35% [SUFFICIENT], 55% [SUBQ], and 10%

`[NO_RETRIEVAL]`. The answering dataset incorporates both no-retrieval and sufficient cases from multiple sources. The size of the answering dataset exceeds the combined total of no-retrieval and sufficient cases from the planning dataset for two reasons: first, we incorporated additional data from ASQA and Arc-Easy datasets; second, we included no-retrieval cases that were initially filtered out in step 1 of our process. This comprehensive approach ensures a robust training set for the answering component.

Our labeling approach ensures that models learn both the iterative nature of complex question answering and the importance of structured knowledge representation. The complementary training objectives help develop robust reasoning capabilities while maintaining retrievability of supporting evidence.

## C.4 DATASET STATISTICS

|  |  | Planning Data | Answering Data |
|---|---|---|---|
| # Queries |  | 129,902 | 78,164 (w/ 3,897 ASQA & 2,147 Arc-Easy) |
| # Input Tokens | (Mean, Median, Max) | (338, 301, 1,910) | (475, 466, 2,214) |
| # Output Tokens | (Mean, Median, Max) | (13, 12, 62) | (13, 4, 2,332) |
| # Subqueries | (Min, Mean, Max) | (0, 0.8, 5) | (0, 1.2, 6) |
| # [SUFFICIENT] |  | 45,722 | – |
| # [SUBQ] |  | 71,676 | – |
| # [NO_RETRIEVAL] |  | 12,504 | – |
| # [SUBQ] Tokens | (Min, Mean, Max) | (6, 18.3, 62) | – |
| # Nodes | (Mean, Median, Max) | (11.6, 11, 56) | (11.7, 11, 84) |
| # Edges | (Mean, Median, Max) | (21.2, 20, 116) | (21.3, 20, 160) |

Table 4: Statistics of our constructed HotpotQA-SUBQ dataset.

## C.5 DATA EXAMPLES

"**input**": "
You are a planner to determine if the question can be answered with current information and output the appropriate label as well as the subquery if needed.
Output [NO_RETRIEVAL] if the question can be directly answered with the question itself without any retrieval.
Output [SUBQ] with an subquery for retrieval if still needs a subquery.
Output [SUFFICIENT] if the question can be answered with the provided information.

**Question**: Given a chat history separated by new lines, generates an informative, knowledgeable and engaging response. ## Input:\n\nI love pizza. While it's basically just cheese and bread you can top a pizza with vegetables, meat etc. You can even make it without cheese!\nPizza is the greatest food ever! I like the New York style.\nI do too. I like that the crust is only thick and crisp at the edge, but soft and thin in the middle so its toppings can be folded in half.\nAbsolutely! I am not that big of a fan of Chicago deep dish though.",

"**label**": "**[NO_RETRIEVAL]**"

Figure 8: Training Data Example (Planning – [NO_RETRIEVAL])

"**input**": "
You are a planner to determine if the question can be answered with current information and output the appropriate label as well as the subquery if needed.
Output [NO_RETRIEVAL] if the question can be directly answered with the question itself without any retrieval.
Output [SUBQ] with an subquery for retrieval if still needs a subquery.
Output [SUFFICIENT] if the question can be answered with the provided information.

**[SUBQ]** Where is The Pick Motor Company Limited located?
**Retrieved Graph Information**: ['(S> Pick Motor Company Limited| P> Alias| O> New Pick Motor Company)', '(S> Pick Motor Company Limited| P> Location| O> Stamford, Lincolnshire)', '(S> Pick Motor Company Limited| P> Operational period| O> 1899-1925)', '(S> Pick Motor Company Limited| P> Industry| O> Motor vehicle manufacturing)']

**Question**: The Pick Motor Company Limited is located in a town on which river ?",

"**label**": "**[SUBQ] Which river is Stamford located on?**"

Figure 9: Training Data Example (Planning – [SUBQ])

"**input**": "
You are a planner to determine if the question can be answered with current information and output the appropriate label as well as the subquery if needed.
Output [NO_RETRIEVAL] if the question can be directly answered with the question itself without any retrieval.
Output [SUBQ] with an subquery for retrieval if still needs a subquery.
Output [SUFFICIENT] if the question can be answered with the provided information.

[SUBQ] What gaming control board in Ohio is Martin R. Hoke a member of?
**Retrieved Graph Information**: ['(S> Martin R. Hoke| P> Former position| O> Member of the United States House of Representatives)', '(S> Martin R. Hoke| P> Birth date| O> May 18, 1952)', '(S> Martin R. Hoke| P> Occupation| O> Politician)', '(S> Martin R. Hoke| P> State| O> Ohio)', '(S> Martin R. Hoke| P> Nationality| O> American)', '(S> Martin R. Hoke| P> Party| O> Republican)', '(S> Martin R. Hoke| P> Member of| O> Ohio Casino Control Commission)', '(S> Martin R. Hoke| P> Born| O> 1952)']

[SUBQ] What gaming control board provides oversight of Ohio's casinos?
**Retrieved Graph Information**: [\"(S> Ohio Casino Control Commission| P> Function| O> Provides oversight of the state's casinos)\", '(S> Ohio Casino Control Commission| P> Location| O> Ohio)', '(S> Ohio Casino Control Commission| P> Type| O> Gaming control board)', '(S> Ohio Casino Control Commission| P> Abbreviation| O> OCCC)']

**Question:** Martin R. Hoke, is an American Republican politician, member of which gaming control board in Ohio that provides oversight of the state's casinos?
",

"**label**": "**[SUFFICIENT]**"

Figure 10: Training Data Example (Planning – [SUFFICIENT])

"**input**": "
You are a answerer given a question and retrieved graph information.
Each [SUBQ] is a subquery we generated through reasoning for the question. The retrieved graph information follows each [SUBQ] is relevant graph information we retrieved to answer the subquery.
[NO_RETRIEVAL] means the question can be answered with the question itself without any retrieval.
The main question starts with \"Question: \". Please answer the question, with subqueries and retrieved graph information if they are helpful.

[NO_RETRIEVAL]
**Question**: Which person won the Nobel Prize in Literature in 1961, Ivo Andri or Nicholas Pileggi?",

"**label**": "Ivo Andri"

Figure 11: Training Data Example (Answering – [NO_RETRIEVAL])

"**input**": "
You are a answerer given a question and retrieved graph information.
Each [SUBQ] is a subquery we generated through reasoning for the question. The retrieved graph information follows each [SUBQ] is relevant graph information we retrieved to answer the subquery.
[NO_RETRIEVAL] means the question can be answered with the question itself without any retrieval.
The main question starts with \"Question: \". Please answer the question, with subqueries and retrieved graph information if they are helpful.

[SUBQ] What type of athlete is Darold Williamson?
**Retrieved Graph Information**: ['(S> Darold Williamson| P> Nationality| O> American)', '(S> Darold Williamson| P> Birth date| O> February 19, 1983)', '(S> Darold Williamson| P> Occupation| O> Track athlete)']

[SUBQ] What specific skills are included in track and field events?
**Retrieved Graph Information:** ['(S> Athletics| P> Includes| O> Road running)', '(S> Track and field| P> Includes| O> Throwing)', '(S> Track and field| P> Includes| O> Jumping)', '(S> Athletics| P> Includes| O> Cross country running)', '(S> Track and field| P> Categorised under| O> Athletics)', '(S> Track and field| P> Includes| O> Running)', '(S> Track and field| P> Venue| O> Stadium with an oval running track enclosing a grass field)', '(S> Athletics| P> Includes| O> Track and field)', '(S> Athletics| P> Includes| O> Race walking)', '(S> Track and field| P> Based on skills of| O> Running)', '(S> Track and field| P> Based on skills of| O> Jumping)', '(S> Track and field| P> Based on skills of| O> Throwing)']

**Question**: Darold Williamson is an athlete in what running and jumping sport?",

"**label**": "**Track and field**"

Figure 12: Training Data Example (Answering – [SUFFICIENT])

# D    TRAINING DETAILS

## D.1    TEXT-TO-TRIPLES MODEL TRAINING

For our text-to-triple conversion model, we fine-tuned a Flan-T5-Large model (Chung et al., 2024) and a LLaMA-3.2-3B-Instruct model (Grattafiori et al., 2024) to transform raw text passages into structured knowledge triples. The model processes input text sequences up to 512 tokens in length and generates structured triples in a standardized format "(S > subject| P> predicate| O> object)".

Our training dataset WikiOFGraph (Kim et al., 2024a), curated by ChatGPT, comprises 5,851,776 text-triple pairs, with an additional 5,000 samples reserved for validation. Each training instance consists of a natural language text passage paired with its corresponding set of comma-separated triples. For example:

---

**Text:**
"William Gerald Standridge (November 27, 1953 – April 12, 2014) was an American stock car racing driver. He was a competitor in the NASCAR Winston Cup Series and Busch Series."

**Triples:**
(S> William gerald standridge| P> Nationality| O> American),
(S> William gerald standridge| P> Occupation| O> Stock car racing driver),
(S> William gerald standridge| P> Competitor| O> Busch series),
(S> William gerald standridge| P> Competitor| O> Nascar winston cup series),
(S> William gerald standridge| P> Birth date| O> November 27, 1953),
(S> William gerald standridge| P> Death date| O> April 12, 2014)

---

Figure 13: Example of WikiOFGraph data

We implemented the training using the AdamW optimizer with a learning rate of 2e-5, incorporating linear warmup and decay schedules. The training process ran for 500,000 steps with a batch size of 32 per GPU and gradient accumulation over 4 steps. To optimize training efficiency and memory usage, we employed mixed-precision training using bfloat16 and applied weight decay at 0.01. The maximum source and target sequence lengths were set to 512 and 256 tokens respectively.

For evaluation, we primarily relied on ROUGE-1, ROUGE-2, and ROUGE-L scores to assess the quality of triple generation. We supplemented these metrics with custom triple matching accuracy measures that consider subject matching, predicate normalization, and object entity alignment. Validation metrics were computed at 5,000-step intervals throughout training.

The training curves for various metrics are shown in Figure 14, demonstrating steady improvement in the model's ability to extract structured knowledge from text. The training of this model was conducted on eight NVIDIA RTX 6000 with 48GB memory.

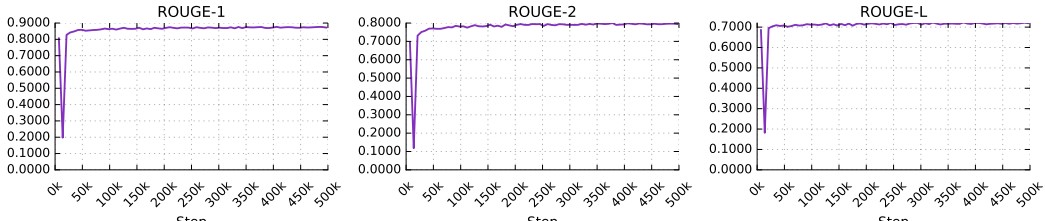

Figure 14: Text-to-Triple Model (base model: Flan-T5-Large) Training Curves.

### D.2 MULTITASK TRAINING OF PLANNING AND ANSWERING CAPABILITIES

We employ Graph LLM (Perozzi et al., 2024; He et al., 2024) as our foundational model architecture and utilize our constructed HotpotQA-SUBQ dataset to train a unified model capable of performing both graph-conditioned action planning and answer generation through multitask learning.

Since both tasks share an identical architectural foundation, as illustrated in Figure **??**, we differentiate their functionality through specialized instruction sets. For Planning, we employ the following instruction template:

> You are a planner to determine if the question can be answered with current information and output the appropriate label as well as the subquery if needed.
> Output [NO_RETRIEVAL] if the question can be directly answered with the question itself without any retrieval.
> Output [SUBQ] with an subquery for retrieval if still needs a subquery.
> Output [SUFFICIENT] if the question can be answered with the provided information.

Figure 15: Instruction `INST_Plan` for Planning.

For the Answering, we utilize this distinct instruction set:

> You are an answerer given a question and retrieved graph information.
> Each [SUBQ] is a subquery we generated through reasoning for the question. The retrieved graph information follows each [SUBQ] is relevant graph information we retrieved to answer the subquery.
> [NO_RETRIEVAL] means the question can be answered with the question itself without any retrieval.
> The main question starts with "Question: ". Please answer the question, with subqueries and retrieved graph information if they are helpful.

Figure 16: Instruction `INST_Ans` for Answering.

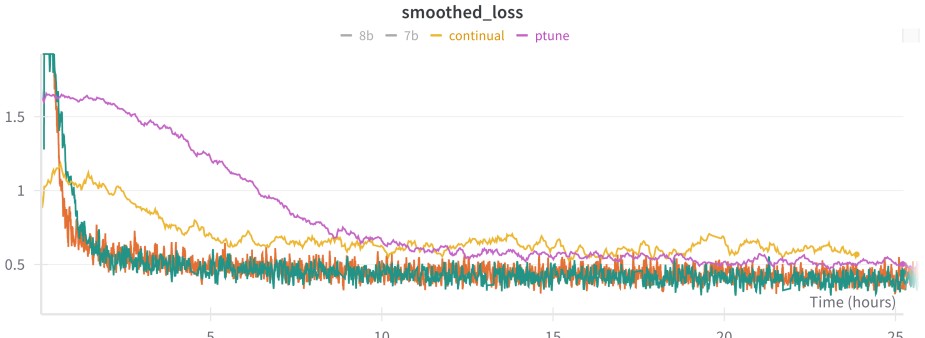

Figure 17: Training loss comparison over one epoch. The plot compares RAS-7B (green) and RAS-8B (orange) training trajectories. Two additional RAS-7B variants are shown: "continual" (yellow): a continual learning approach where Answering training precedes Planning training, and "ptune" (purple): a parameter-efficient variant that only tunes graph tokens without LoRA. The lower and more stable loss curves of the standard RAS variants demonstrate the effectiveness of joint training with LoRA. We use the last 100 steps for loss smoothing.

For graph encoding, we implement Graph Transformer (Shi et al., 2020), selected for its robust capability in handling non-fully-connected graphs—a common characteristic of text-extracted triples. Our base language models comprise LLaMA-2-7B and LLaMA-3-8B, chosen both to maintain

consistency with previous research (Asai et al., 2023; Lyu et al., 2024b) and to investigate our framework's performance scaling across different model capacities.

Our implementation of GraphLLM differs from G-Retriever (He et al., 2024) primarily due to the distinct nature of our graph structures. While G-Retriever operates on single interconnected graphs, our framework processes multiple potentially disconnected subgraphs, each corresponding to different subqueries. To address this architectural difference, we adopt a sequential encoding strategy: rather than encoding the entire graph at once, we process each subgraph individually using Graph Transformer, followed by mean pooling across all subgraph embeddings to produce the final encoded representation.

The training process utilizes 4 NVIDIA RTX 6000 Ada Generation GPUs, each with 48GB memory. We train all models for 2 epochs using a batch size of 2 and gradient accumulation steps of 2, implementing a peak learning rate of 1e-5 with a 0.15 warmup ratio and 0.01 decay rate. The maximum sequence lengths are set to 300 tokens for generation and 2,500 tokens for input, with training conducted in BFloat16 precision.

# E    EVALUATION DATASETS & METRICS

## E.1    TEST DATASETS

We evaluate RAS on diverse benchmark datasets spanning short-form QA, closed-set tasks, and long-form generation in the zero-shot setting, aligning with (Asai et al., 2023; Lyu et al., 2024b). Below we describe each dataset:

**Short-form Generation Datasets:**

- **TriviaQA-unfiltered (TQA)** (Joshi et al., 2017): A large-scale QA dataset containing 11,313 question-answer pairs in our test set. The questions are sourced from trivia enthusiasts and cover diverse topics.
- **2WikiMultiHopQA (2WQA)** (Ho et al., 2020): A multi-hop question answering dataset (with 12,576 samples in test set) that requires models to combine information from multiple Wikipedia articles to answer questions.
- **PopQA** (Mallen et al., 2022): A dataset focusing on questions about long-tail entities, containing 1,399 queries where the monthly Wikipedia page views are less than 100. These questions test models' ability to handle queries about less common entities.

**Closed-set Tasks:**

- **PubHealth** (Pub) (Zhang et al., 2023a): A dataset for verifying health-related claims. Models must classify statements as "true" or "false" based on scientific evidence. The dataset contains 987 test samples.
- **ARC-Challenge** (ARC) (Clark et al., 2018): A multiple-choice science question dataset designed to be challenging for models, requiring multi-hop reasoning and background knowledge. The dataset contains 1,172 test samples.

**Long-form Generation Datasets:**

- **ASQA** (Stelmakh et al., 2022): A long-form question answering dataset that requires models to generate comprehensive answers with proper citation of sources. Models must balance information completeness with factual accuracy. The dataset contains 948 test samples.
- **ELI5** (Fan et al., 2019): A dataset derived from the "Explain Like I'm Five" subreddit, containing questions seeking straightforward explanations of complex topics. The dataset contains 1,000 test samples.

## E.2 EVALUATION METRICS

We employ different evaluation metrics appropriate for each task category:

**Short-form Generation**:

- For PopQA and TriviaQA, we use the "golden match" metric (Asai et al., 2023; Min et al., 2019; Guu et al., 2020)[2], where a prediction $p$ is considered correct if it contains any normalized version of the ground truth answers $G = \{g_1, ..., g_n\}$:

$$\text{match}(p, G) = \begin{cases} 1 & \text{if } \exists g \in G : \text{norm}(g) \subseteq \text{norm}(p) \\ 0 & \text{otherwise} \end{cases} \tag{9}$$

  where $\text{norm}(\cdot)$ normalizes text by lowercasing, removing articles and punctuation.

- For 2WikiMultiHopQA, we follow RPG (Lyu et al., 2024b)[3] to use token-level F1 score between prediction $p$ and ground truth $g$:

$$\text{F1}(p, g) = 2 \cdot \frac{\text{precision} \cdot \text{recall}}{\text{precision} + \text{recall}} \tag{10}$$

  where precision and recall are computed over normalized token overlap:

$$\text{precision} = \frac{|\text{tokens}(p) \cap \text{tokens}(g)|}{|\text{tokens}(p)|} \tag{11}$$

$$\text{recall} = \frac{|\text{tokens}(p) \cap \text{tokens}(g)|}{|\text{tokens}(g)|} \tag{12}$$

**Closed-set Tasks**: For both PubHealth and ARC-Challenge, we use accuracy (Asai et al., 2023), computed as:

$$\text{accuracy} = \frac{|\{i : \text{norm}(p_i) = \text{norm}(g_i)\}|}{N} \times 100 \tag{13}$$

where $N$ is the total number of examples.

**Long-form Generation**: For ASQA and ELI5, we use multiple metrics[4]:

- ROUGE-L score to measure the longest common subsequence between prediction and reference (Lin, 2004)
- MAUVE score (Pillutla et al., 2021) to test the generation fluency by comparing the distribution of generated text against human references

All scores are reported as percentages, multiplied by 100.

---

[2]Implementation aligned with https://github.com/AkariAsai/self-rag/blob/main/retrieval_lm/metrics.py

[3]Implementation aligned with https://github.com/haruhi-sudo/RPG/blob/main/retriever/src/evaluation.py

[4]Implementation aligned with https://github.com/princeton-nlp/ALCE/blob/main/eval.py

# F INFERENCE & EVALUATION DETAILS

We evaluated RAS using both closed-source and open-source language models. For closed-source evaluation, we used Claude-3.5-Sonnet (Sonnet-3.5) as our base model. For open-source evaluation, we tested with both LLaMA-2-7B and LLaMA-3-8B, as shown in the performance table (Table 1).

For the open-source models, we first fine-tuned the GraphLLM architecture on our HotpotQA-SUBQ dataset (see Section D.2). We then conducted zero-shot knowledge-intensive generation tests. With Claude-3.5-Sonnet, we used few-shot prompting for both action planning and answer generation phases. Below, we detail our approach for each model type.

## F.1 CLOSED-SOURCE MODEL SETTINGS

**For Text-to-Triple Conversion** (Figure 18), we instruct the model to extract structured knowledge triples following specific formatting rules. The prompt precisely defines the triple format as $(S > subject \mid P > predicate \mid O > object)$ and provides comprehensive guidelines to ensure consistent knowledge representation. Key rules include extracting maximal meaningful relationships, maintaining original entity casing, avoiding pronoun usage, and ensuring clear predicate specification. The example demonstrates the conversion of a biographical text into structured triples, showing how complex information about an individual's life, career, and temporal details can be systematically decomposed into atomic facts. This standardized format enables reliable knowledge accumulation and reasoning in subsequent stages of the RAS framework.

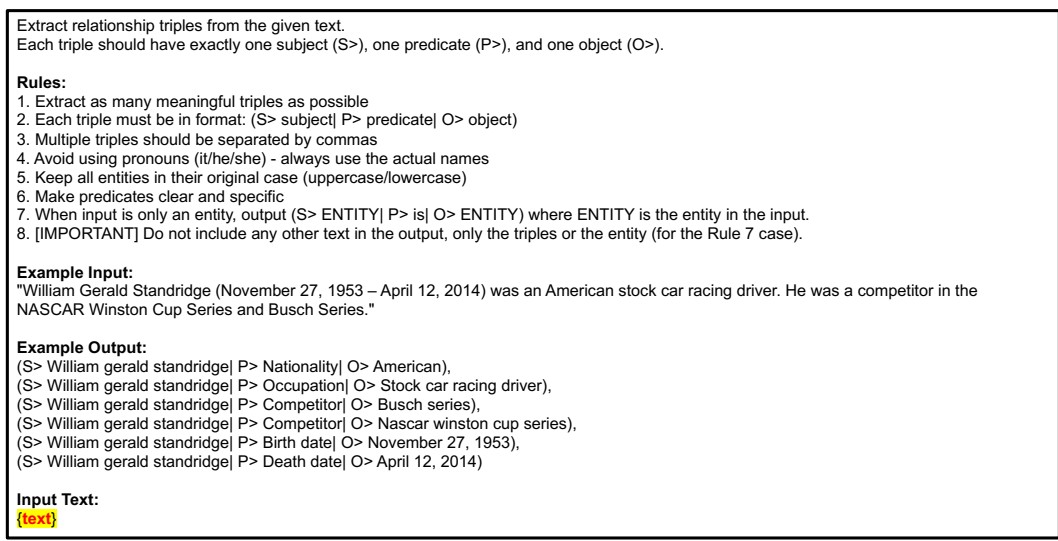

Figure 18: Few-shot prompt for text-to-triples transformation with closed-source LLM.

**For Planning** (Figure 19), we design a comprehensive prompt that guides the model in making strategic decisions about information retrieval needs. The prompt instructs the model to output one of three labels based on careful analysis of the current knowledge state: (1) `[NO_RETRIEVAL]` when the question can be answered directly, either due to the model's inherent knowledge or the question's nature, (2) `[SUBQ]` accompanied by a focused subquery when additional information is needed, or (3) `[SUFFICIENT]` when the accumulated knowledge is adequate to answer the question. The prompt includes diverse examples demonstrating different scenarios:

(1) Generating follow-up queries based on partial information (2) Creating new queries when relevant information is missing (3) Recognizing when accumulated information is sufficient (4) Identifying questions that don't require external knowledge (5) Handling common knowledge questions

A key feature of the prompt is its emphasis on query efficiency - it explicitly prohibits generating redundant subqueries that might retrieve already-known information. This design helps maintain the system's efficiency while ensuring comprehensive knowledge gathering.

You are a planner to determine if the question can be answered with current information (Subquery [PREV_SUBQ] and retrieved graph information [PREV_GRAPH_INFO]) and output the appropriate label as well as the subquery if needed.
Output [NO_RETRIEVAL] if the question can be directly answered with the question itself without any retrieval. You are expected to output [NO_RETRIEVAL] either if you believe an LLM is knowledgeable enough to answer the question, or if you believe the question type is not suitable for retrieval.
Output [SUBQ] with an subquery for retrieval if still needs a subquery. Do not make an similar subquery that has been made before ([PREV_SUBQ]), as it is very likely to retrieve the same information.
Output [SUFFICIENT] if the question can be answered with the provided information.
The main question starts with "Question: ".

**Examples:**
---------------------------------------------------------------------------------------------
Example 1 (Use the information in [PREV_GRAPH_INFO] to further generate a new subquery):
Input:
[PREV_SUBQ] Where is The Pick Motor Company Limited located?
[PREV_GRAPH_INFO] ['(S> Pick Motor Company Limited| P> Alias| O> New Pick Motor Company)', '(S> Pick Motor Company Limited| P> Location| O> Stamford, Lincolnshire)', '(S> Pick Motor Company Limited| P> Operational period| O> 1899-1925)', '(S> Pick Motor Company Limited| P> Industry| O> Motor vehicle manufacturing)']
Question: The Pick Motor Company Limited is located in a town on which river ?,

Output:
[SUBQ] Which river is Stamford located on?

---------------------------------------------------------------------------------------------
Example 2 (No relevant information found in [PREV_GRAPH_INFO]):
Input:
[PREV_SUBQ] What medals did Michael Johnson win in the 1996 Olympics?
[PREV_GRAPH_INFO] ['(S> Michael Johnson| P> Nationality| O> American)', '(S> Michael Johnson| P> Birth date| O> September 13, 1967)', '(S> Michael Johnson| P> Sport| O> Track and field)', '(S> Michael Johnson| P> Team| O> United States Olympic team)']
Question: What was Michael Johnson's winning time in the 400m at the 1996 Olympics?

Output:
[SUBQ] What records or times did Michael Johnson set in the 400m at the 1996 Olympic Games?

---------------------------------------------------------------------------------------------
Example 3 (The current information is sufficient to answer the question):
Input:
[PREV_SUBQ] What gaming control board in Ohio is Martin R. Hoke a member of?
[PREV_GRAPH_INFO] ['(S> Martin R. Hoke| P> Former position| O> Member of the United States House of Representatives)', '(S> Martin R. Hoke| P> Birth date| O> May 18, 1952)', '(S> Martin R. Hoke| P> Occupation| O> Politician)', '(S> Martin R. Hoke| P> State| O> Ohio)', '(S> Martin R. Hoke| P> Nationality| O> American)', '(S> Martin R. Hoke| P> Party| O> Republican)', '(S> Martin R. Hoke| P> Member of| O> Ohio Casino Control Commission)', '(S> Martin R. Hoke| P> Born| O> 1952)']
[PREV_SUBQ] What gaming control board provides oversight of Ohio's casinos?
[PREV_GRAPH_INFO] ['"(S> Ohio Casino Control Commission| P> Function| O> Provides oversight of the state's casinos)\"', '(S> Ohio Casino Control Commission| P> Location| O> Ohio)', '(S> Ohio Casino Control Commission| P> Type| O> Gaming control board)', '(S> Ohio Casino Control Commission| P> Abbreviation| O> OCCC)']
Question: Martin R. Hoke, is an American Republican politician, member of which gaming control board in Ohio that provides oversight of the state's casinos?

Output:
[SUFFICIENT]

---------------------------------------------------------------------------------------------
Example 4 (The question is not suitable for retrieval):
Input:
Given a chat history separated by new lines, generates an informative, knowledgeable and engaging response.
##Input:
I love pizza. While it's basically just cheese and bread you can top a pizza with vegetables, meat etc. You can even make it without cheese! Pizza is the greatest food ever! I like the New York style.
I do too. I like that the crust is only thick and crisp at the edge, but soft and thin in the middle so its toppings can be folded in half.
Absolutely! I am not that big of a fan of Chicago deep dish though

Output:
[NO_RETRIEVAL]

---------------------------------------------------------------------------------------------
Example 5 (You are knowledgeable enough to answer the question):
Input:
What is the capital of the United States?

Output:
[NO_RETRIEVAL]

---------------------------------------------------------------------------------------------
[VERY IMPORTANT] Please only either output (1) [NO_RETRIEVAL] or (2) [SUBQ] with an concrete subquery for retrieval, or (3) [SUFFICIENT] if the question can be answered with the provided information.
[VERY IMPORTANT] Do not output any other text. DO NOT make an identical subquery [SUBQ] that has been made before ([PREV_SUBQ])!

Now, your turn:
**Input:**
{planner_intput}

**Output:**

Figure 19: Few-shot prompt for planning with closed-source LLM.

**For Answering** (Figure 20), we design a prompt that focuses on synthesizing precise answers from structured knowledge graphs. The prompt emphasizes selective use of retrieved information - instructing the model to utilize subqueries and graph information only when relevant to the question at hand. A distinctive feature of this prompt is its requirement for definitive answers even under uncertainty or incomplete information, ensuring the model always provides a response.

The prompt includes carefully selected examples demonstrating two key scenarios: direct fact retrieval (Nobel Prize winner) and complex reasoning across multiple knowledge pieces (athlete's sport classification). These examples illustrate how to effectively combine information from multiple subqueries and their associated graph information to construct accurate answers. The prompt strictly enforces concise answer generation, requiring only the essential information without additional explanation or commentary.

```
You are a answerer given a question and retrieved graph information.
Each [SUBQ] is a subquery we generated through reasoning for the question. The retrieved graph information follows each [SUBQ] is relevant
graph information we retrieved to answer the subquery.
The main question starts with "Question: ". Please answer the question, with subqueries and retrieved graph information if they are helpful (do
not use them if they are not helpful).
You must answer the question, even if there's no enough information to answer the question, or you are not sure about the answer.

Examples:
-------------------------------------------------------------------------------------
Example 1:
Input:
Question: Which person won the Nobel Prize in Literature in 1961, Ivo Andri or Nicholas Pileggi?",

Output:
Ivo Andri

-------------------------------------------------------------------------------------
Example 2:
[SUBQ] What type of athlete is Darold Williamson?
Retrieved Graph Information: ['(S> Darold Williamson| P> Nationality| O> American)', '(S> Darold Williamson| P> Birth date| O> February 19,
1983)', '(S> Darold Williamson| P> Occupation| O> Track athlete)']
[SUBQ] What specific skills are included in track and field events?
Retrieved Graph Information: ['(S> Athletics| P> Includes| O> Road running)', '(S> Track and field| P> Includes| O> Throwing)', '(S> Track and
field| P> Includes| O> Jumping)', '(S> Athletics| P> Includes| O> Cross country running)', '(S> Track and field| P> Categorised under| O>
Athletics)', '(S> Track and field| P> Includes| O> Running)', '(S> Track and field| P> Venue| O> Stadium with an oval running track enclosing a
grass field)', '(S> Athletics| P> Includes| O> Track and field)', '(S> Athletics| P> Includes| O> Race walking)', '(S> Track and field| P> Based on
skills of| O> Running)', '(S> Track and field| P> Based on skills of| O> Jumping)', '(S> Track and field| P> Based on skills of| O> Throwing)']
Question: Darold Williamson is an athlete in what running and jumping sport?",

Output:
Track and field

-------------------------------------------------------------------------------------
[VERY IMPORTANT] Please only output the answer to the question.
[VERY IMPORTANT] Do not output any other text.

Now, your turn:
Input:
{answerer_input}

Output:
```

Figure 20: Few-shot prompt for answering with closed-source LLM.

In addition, for ASQA and ELI5 with closed-source models (RAS$_{Sonnet-3.5}$ and all the other baselines (e.g., SuRe) using Sonnet-3.5), we conduct few-shot inference with two in-context learning demonstrations, aligning with previous study's (Gao et al., 2023) implementation.[56]

---

[5] https://github.com/princeton-nlp/ALCE/blob/main/run.py
[6] Exemplars can be found at https://github.com/princeton-nlp/ALCE/tree/main/prompts

## F.2 OPEN-SOURCE MODEL SETTINGS

For inference with open-source model, we use our GraphLLM trained by Hotpot-SUBQ (see Appendix D.2), we use 8 NVIDIA RTX 6000 Ada Generation with 48GB memory and CUDA version 12.4. We use Python 3.10, PyTorch 2.1.2, and torch-geometric 2.6.1 throughout all experiments.

For more efficient text retrieval, we split the dense index of the corpus into eight splits, and load them on eight GPUs with faiss-gpu 1.7.2.[7]

We set maximum new tokens as **100 for PopQA, TriviaQA, and 2WikiMultihopQA**, as **50 for PubHealth and ARC-Challenge**, and as **300 for ASQA and ELI5**, aligning with previous study (Asai et al., 2023; Lyu et al., 2024b). All the generation are configured with **batch size of 1**.

As we always set the main question as the first subquery, we pre-extract the triples from the context at the first iteration using our text-to-triple model for batch triple extraction.

We use exact same instructions for planning and answering as we used in training phase, as shown in Figures 15 and 16.

## F.3 INSTRUCTIONS

We use the following task-specific instructions for zero-shot inference:

| Dataset | Instruction |
|---|---|
| ARC-C (baseline) | Given four answer candidates, A, B, C and D, choose the best answer choice. Please answer with the capitalized alphabet only, without adding any extra phrase or period. |
| ARC-C | Which is true? Output A, B, C, or D. |
| PubHealth (baseline) | Is the following statement correct or not? Say true if it's correct; otherwise, say false. Don't capitalize or add periods, just say "true" or "false". |
| PubHealth | Is statement 'true' or 'false'? Only output 'true' or 'false'. |
| ASQA (baseline) | Instruction: Write an ACCURATE, ENGAGING, and CONCISE answer for the given question using the retrieved graph information (some of which might be irrelevant). Use an unbiased and journalistic tone. |
| ASQA | Answer the following question. The question may be ambiguous and have multiple correct answers, and in that case, you have to provide a long-form answer including all correct answers. [Long Form] |
| ELI5 (baseline) | Instruction: Write an ACCURATE, ENGAGING, and CONCISE answer for the given question using the retrieved graph information (some of which might be irrelevant). Use an unbiased and journalistic tone. |
| ELI5 | Provide a paragraph-length response using simple words to answer the following question. [Long Form] |

Table 5: Task-specific instructions. For short-form QA, we do not use instructions and use the original questions only.

## G COST & EFFICIENCY COMPARISON WITH OTHER GRAPH-BASED RAG METHODS

**Static vs. Dynamic Structuring.** Static corpus-level graph methods such as GraphRAG (Edge et al., 2024) and HippoRAG (Gutiérrez et al., 2024) require building and maintaining large knowledge graphs offline. This involves millions of LLM calls for entity/relation extraction and summarization, plus heavy clustering and embedding computation. At the scale of Wikipedia 2018 (roughly 5M articles, ~3B tokens), these pipelines incur substantial costs, days of GPU/CPU compute, and ≥50 GB storage overhead. In contrast, our RAS constructs **question-specific knowledge graphs dynamically at inference**, eliminating expensive offline indexing. The only extra step is lightweight triple extraction from retrieved passages, which we implement with a small LLaMA-3B model accelerated by vLLM (Kwon et al., 2023), achieving throughput of nearly 5K tokens/s (Table 3).

**Cost Estimation Methodology.** We estimate costs based on empirical data from Microsoft's GraphRAG analysis [8] and scale to Wikipedia 2018. Cost estimates assume current API pricing and may vary significantly based on implementation details, model selection, and optimization strategies.

---

[7]All the Wikipedia dumps (2018, 2020) we used are downloaded from https://github.com/facebookresearch/atlas

[8]https://techcommunity.microsoft.com/blog/azure-ai-foundry-blog/graphrag-costs-explained-what-you-need-to-know/4207978

**Computation of Indexing Cost.** We estimate costs using the following assumptions:

- **Corpus size:** Wikipedia 2018 ≈ 3B tokens, or ≈5M articles with ∼600 tokens/article.
- **GraphRAG pipeline:** Based on Microsoft's empirical analysis showing $0.000011 per word for GPT-4o-mini.
    - For 3B tokens (≈2.25B words), base processing cost: $25k–$30k
    - Entity/relation extraction with few-shot prompting increases token usage by 3–5×: $75k–$150k
    - Community summarization for ∼10k communities at $0.02 per community: $200–$500
    - Additional overhead (clustering, embeddings, storage): $10k–$25k
    - **Total GraphRAG:** $85k–$175k for GPT-4o-mini; $250k–$500k for GPT-4o
- **HippoRAG pipeline:** More efficient extraction but still corpus-wide processing.
    - Entity extraction: 3B tokens at reduced complexity ≈ $15k–$30k
    - Synonym linking and embedding overhead: $5k–$15k
    - Graph construction and indexing: $5k–$10k
    - **Total HippoRAG:** $25k–$55k
- **RAS pipeline:** No offline corpus-wide indexing. Triple extraction uses local LLaMA-3B model.
    - Offline indexing cost: $0 (no corpus preprocessing required)
    - Runtime triple extraction: Local model inference (negligible marginal cost)
    - **Total RAS:** $0 offline costs

**Cost Sensitivity Analysis.** Costs vary significantly based on:

- **Model selection:** GPT-4o costs ≈3× more than GPT-4o-mini
- **Chunk size:** Smaller chunks (300 vs 1200 tokens) can reduce costs by 30–50%
- **Prompt optimization:** Well-tuned prompts can reduce token usage by 20–40%
- **Processing parameters:** Conservative estimates assume non-optimized settings

**Quantitative Comparison.** Table 6 summarizes estimated costs with uncertainty ranges.

| Method | Offline LLM Cost | Compute Time | Storage | Ready for RAG? |
|---|---|---|---|---|
| GraphRAG | $85k–$500k (model-dependent) | >7 days (distributed) | 50–70 GB | Yes (community reports) |
| HippoRAG | $25k–$55k (optimized extraction) | >3 days (cluster) | 30–50 GB | Yes (node-passage matrix) |
| **RAS (ours)** | **Negligible** | Inference-time only | Small (triple cache) | Yes (dynamic graph) |

Table 6: Estimated cost comparison for building knowledge graph indexes at Wikipedia 2018 scale. Ranges reflect model choice, optimization level, and implementation variations.

**Efficiency Visualization.** We visualize the contrast in *offline indexing cost* ranges. RAS requires no precomputation, while GraphRAG and HippoRAG incur substantial upfront costs.

**Takeaway.** RAS achieves the structured reasoning benefits of graphs *without* any offline indexing costs, compared to GraphRAG ($85k–$500k) and HippoRAG ($25k–$55k). By eliminating upfront investment requirements and enabling pay-per-query scaling through local model inference, RAS makes graph-enhanced RAG accessible for scenarios where large-scale preprocessing is impractical or cost-prohibitive.

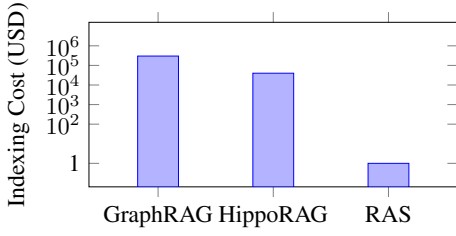

Figure 21: Estimated offline indexing cost ranges at Wikipedia scale (log scale).

## H  SUPPLEMENTARY RESULTS AND ANALYSIS

### Higher Performance with More Contexts?

We further examine the performance of Sonnet-3.5 when increasing the number of retrieved documents. Table 7 compares Sonnet-3.5 under different retrieval depths with RAS-Sonnet.

| Method | 2WikiMultihopQA (F1) | PopQA (Acc) | PubHealth (Acc) | ARC (Acc) |
|---|---|---|---|---|
| Sonnet-3.5 (#docs=5) | 53.7 | 57.3 | 53.9 | 87.1 |
| Sonnet-3.5 (#docs=10) | 55.1 | 58.5 | 56.6 | 87.9 |
| Sonnet-3.5 (#docs=15) | 55.4 | 58.0 | 56.9 | 86.9 |
| **RAS (Sonnet)** | **57.7** | **62.3** | **71.4** | **93.8** |

Table 7: Performance comparison between Sonnet-3.5 under varying context sizes and RAS-Sonnet.

Increasing retrieval depth yields diminishing returns for Sonnet-3.5, while RAS-Sonnet achieves significantly higher performance with structured, targeted context. This highlights the limitations of passive context scaling and the benefits of RAS's organized retrieval strategy.

### Error Analysis on Graph Construction

We present two representative cases where the quality of retrieved structured knowledge significantly impacts final answer generation. These examples are distinct from the qualitative examples shown in Section I and highlight both successful and failed reasoning paths.

**Case 1: Accurate Reasoning with Well-Structured Graph**
**Question:** Who directed the film adaptation of the novel "Tinker Tailor Soldier Spy"?
**Subqueries and Graphs:**

- `[SUBQ] Who directed the film "Tinker Tailor Soldier Spy"?`
  (S> Tinker Tailor Soldier Spy | P> Film adaptation director | O> Tomas Alfredson),
  (S> Tinker Tailor Soldier Spy | P> Release year | O> 2011)

- `[SUBQ] Who is Tomas Alfredson?`
  (S> Tomas Alfredson | P> Occupation | O> Film director)

**Answer:** *Tomas Alfredson* (Correct)
**Analysis:** The subquery identifies the correct adaptation and its director. The structured triple precisely encodes this relation, allowing the model to reason accurately.

**Case 2: Reasoning Failure Due to Spurious or Missing Triples**
**Question:** What invention is Nikola Tesla best known for?
**Subqueries and Graphs:**

- `[SUBQ] What are Nikola Tesla's major inventions?`
  (S> Nikola Tesla | P> Known for | O> Earthquake machine),
  (S> Nikola Tesla | P> Proposed | O> Death ray),
  (S> Nikola Tesla | P> Created | O> Tesla coil)

- `[SUBQ] What is a Tesla coil used for?`
  (S> Tesla coil | P> Used in | O> High-frequency experiments)

**Answer:** *Earthquake machine or death ray* (Incorrect)
**Analysis:** The graph omits the key triple (Nikola Tesla | Invented | Alternating current system), instead surfacing fringe inventions. This misleads the model into emphasizing less impactful or speculative work, leading to an incorrect answer.

These examples reveal that RAS makes reasoning pathways more interpretable by structuring retrieved knowledge explicitly. When failures occur, they can often be traced to identifiable issues in the extracted triples, such as missing key relations or overemphasis on fringe facts, rather than opaque

model behavior. This transparency not only facilitates error analysis but also suggests concrete avenues for improving upstream components like triple extraction or subquery generation, making RAS a flexible foundation for future refinement.

## I QUALITATIVE EXAMPLES

In this section, we show some qualitative examples and running examples of RAS.

---

**Example 1 (PopQA)**

**[SUBQ]** Who was the screenwriter for The Fake?
**Retrieved Graph Information:**
**(S> The Fake | P> Release year | O> 1953),**
**(S> The Fake | P> Genre | O> British crime film),**
**(S> The Fake | P> Director | O> Godfrey Grayson),**
(S> The Fake | P> Stars | O> Dennis O'Keefe),
(S> The Fake | P> Stars | O> Coleen Gray),
(S> The Fake | P> Stars | O> Hugh Williams),
(S> The Fake | P> Plot concerns | O> American detective solving theft of priceless painting),
(S> Priceless painting | P> Stolen from | O> Tate Gallery),
(S> Tate Gallery | P> Located in | O> London)

**[SUBQ]** Who wrote the screenplay for the 1953 British crime film The Fake?
**Retrieved Graph Information:**
(S> Antony Carr | P> Occupation | O> Author),
(S> Antony Carr | P> Genre | O> Crime novels),
(S> Antony Carr | P> Birth year | O> 1916),
(S> Antony Carr | P> Death year | O> 1995),
(S> Antony Carr | P> Nationality | O> English),
(S> Antony Carr | P> Father | O> Robert Carr),
(S> Robert Carr | P> Occupation | O> Singer),
(S> Antony Carr | P> Niece | O> Susannah Carr),
(S> Susannah Carr | P> Occupation | O> Newsreader),
(S> Susannah Carr | P> Nationality | O> Australian)

**[SUBQ]** Who wrote the script or screenplay for the 1953 British crime film The Fake directed by Godfrey Grayson?
**Retrieved Graph Information:**
(S> The Fake | P> Is a | O> British silent drama film),
(S> The Fake | P> Release year | O> 1927),
(S> The Fake | P> Directed by | O> Georg Jacoby),
(S> The Fake | P> Stars | O> Henry Edwards),
(S> The Fake | P> Stars | O> Elga Brink),
(S> The Fake | P> Stars | O> Juliette Compton),
**(S> The Fake | P> Based on | O> Play of the same title),**
**(S> Play of the same title | P> Written by | O> Frederick Lonsdale),**
(S> Play of the same title | P> Release year | O> 1924),
(S> The Fake | P> Made at | O> Twickenham Studios),
(S> Twickenham Studios | P> Located in | O> London)

**Question:** Who was the screenwriter for The Fake?

**Answer:** Frederick Lonsdale

---

Figure 22: An inference example in PopQA dataset.

**Example 2 (TriviaQA)**

**[SUBQ]** Who is the most successful UK solo artist in the USA?
**Retrieved Graph Information:**
(S> Jay Sean | P> Achievement | O> Most successful male UK urban artist in US chart history),
(S> Jay Sean | P> Song reached number one | O> Down),
(S> Jay Sean | P> Chart topped | O> Billboard Hot 100),
(S> Taio Cruz | P> Topped chart | O> US Billboard Hot 100),
(S> Cliff Richard | P> Claimed worldwide sales | O> 260 million),
(S> Cliff Richard | P> Ranking | O> Second highest selling British male solo artist worldwide),
**(S> Elton John | P> Ranking | O> Highest selling British male solo artist worldwide),**
(S> Cliff Richard | P> Backing band | O> The Shadows),
(S> Robbie Williams | P> Chart topper | O> Rock DJ),
(S> Robbie Williams | P> Collaboration with | O> Kylie Minogue)

**[SUBQ]** What are the US chart records or album sales figures for Elton John in the United States?
**Retrieved Graph Information:**
(S> Elton John | P> Plays | O> Piano),
(S> Elton John | P> Performs | O> Vocals),
(S> Dee Murray | P> Plays | O> Bass),
(S> Nigel Olsson | P> Plays | O> Drums),
(S> Davey Johnstone | P> Plays | O> Electric guitar),
(S> Paul Buckmaster | P> Arranges | O> Orchestral arrangement),
(S> Bob Birch | P> Plays | O> Bass guitar),
(S> Charlie Morgan | P> Plays | O> Drums),
(S> Charlie Morgan | P> Plays | O> Percussion),
(S> Guy Babylon | P> Plays | O> Keyboards),
(S> Matt Springer | P> Ranked | O> Elton John's studio albums),
(S> Ultimate Classic Rock | P> Employs | O> Matt Springer),
(S> Daryl Hall & John Oates | P> Nationality | O> American),
(S> Daryl Hall & John Oates | P> Sold albums in US | O> 13 million),
(S> Daryl Hall & John Oates | P> Sold singles in US | O> 6 million)

**Question:** Who is the most successful UK solo artist in the USA?

**Answer: Elton John**

Figure 23: An inference example in TriviaQA dataset.

**Example 3 (ASQA)**

**[SUBQ]** Who sang the song i'm coming out?
**Retrieved Graph Information:**
(S> Diana Ross | P> Sang | O> I'm Coming Out),
(S> Bernard Edwards | P> Wrote | O> I'm Coming Out),
(S> Nile Rodgers | P> Wrote | O> I'm Coming Out),
(S> Bernard Edwards | P> Produced | O> I'm Coming Out),
(S> Nile Rodgers | P> Produced | O> I'm Coming Out),
(S> I'm Coming Out | P> Released on | O> August 22, 1980),
(S> I'm Coming Out | P> Second single from album | O> Diana),
(S> Diana Ross | P> Commissioned | O> Chic),
(S> Stevie J | P> Sampled | O> I'm Coming Out),
(S> The Notorious B.I.G. | P> Used sample in song | O> Mo Money Mo Problems),
(S> Marcia Hines | P> Covered | O> I'm Coming Out)
(S> Keyshia Cole | P> Remade | O> I'm Coming Out),
(S> Iggy Azalea | P> Featured on remake of | O> I'm Coming Out),
(S> Ariana Grande | P> Sampled in song | O> Break Your Heart Right Back),
(S> Hella von Sinnen | P> Sampled | O> I'm Coming Out),
(S> Amerie | P> Covered | O> I'm Coming Out)

**[Long Form] Question:** Who sang the song I'm coming out?

**Ground Truth:** "I'm Coming Out" is a song recorded by American singer Diana Ross. It was written and produced by Chic members Bernard Edwards and Nile Rodgers, and released on August 22, 1980, as the second single from Ross' self-titled tenth album Diana (1980). In 2003, American singer Amerie covered "I'm Coming Out" for the soundtrack to the film "Maid in Manhattan". Her version was released as a single in select European countries and Australia.

**Generation:** "I'm Coming Out" is a song by American singer Diana Ross. It was written by Bernard Edwards and Nile Rodgers and produced by Edwards and Rodgers. The song was released on August 22, 1980 as the second single from Ross's album Diana. The song was sampled by Stevie J in his 2005 song "U Don't Know" and by The Notorious B.I.G. in his 1997 song "Mo Money Mo Problems". Marcia Hines covered the song for her 1981 album Marcia. Keyshia Cole remade the song for her 2007 album Just Like You. In 2014, Iggy Azalea featured on a remake of the song. Ariana Grande sampled the song in her 2015 song "Break Your Heart Right Back".

---

**Example 4 (ELI5)**

[SUBQ] What is a stock 'put' and how does it work?
Retrieved Graph Information:
(S> Put option | P> Is | O> Financial contract),
(S> Put option | P> Gives buyer | O> Right to sell underlying asset),
(S> Put option | P> Has | O> Predetermined price),
(S> Put option | P> Has | O> Expiration date),
(S> Put option | P> Can be associated with | O> Stocks),
(S> Put option | P> Can be associated with | O> Commodities),
(S> Put option buyer | P> Not obligated to | O> Sell at contract price),
(S> Put option | P> Used for | O> Hedging),
(S> Put option buyer | P> Bets on | O> Stock price decline)
(S> Put option | P> Has | O> Defined risk)

**[Long Form] Question**: What is a stock 'put' and how does it work?

**Ground Truth:** A stock put is a type of stock option, that grants you the option to sell at a specified date for a set price. It's effectively a means of protecting stock you own, or for betting against a stock. Let's say you buy an 1 contract for August 140 puts on Apple. That means that the date August options expire, you would be able to sell 100 shares at $140 if you so choose. So if Apple drops to $125 by that time, you'd protect $15/share of gains. This might only cost you $1-2 per share to buy the contract. Or if you simply think the stock will fall, the put will increase in value as the strike price approaches "in the money." So that $1-2/sh. you spend to buy the put option might be worth $10/share if the stock starts dropping and there's greater likelihood that it'll be below the $140 price. Because there is a set expiration date, the values have a relationship to both the underlying stock and the time until the expiration. So a month out, the put option that's $10 in the money ($140 put, stock trading at $130) might cost $20 while the day of the expiration it'd be just about the difference between option value and stock itself.

**Generation:** A put option is a financial contract that gives the buyer the right to sell the underlying asset at a predetermined price and expiration date. The put option buyer is not obligated to sell the asset at the contract price, but they can bet on the stock price decline.

Figure 24: Examples of long-form generation on ASQA and ELI5.

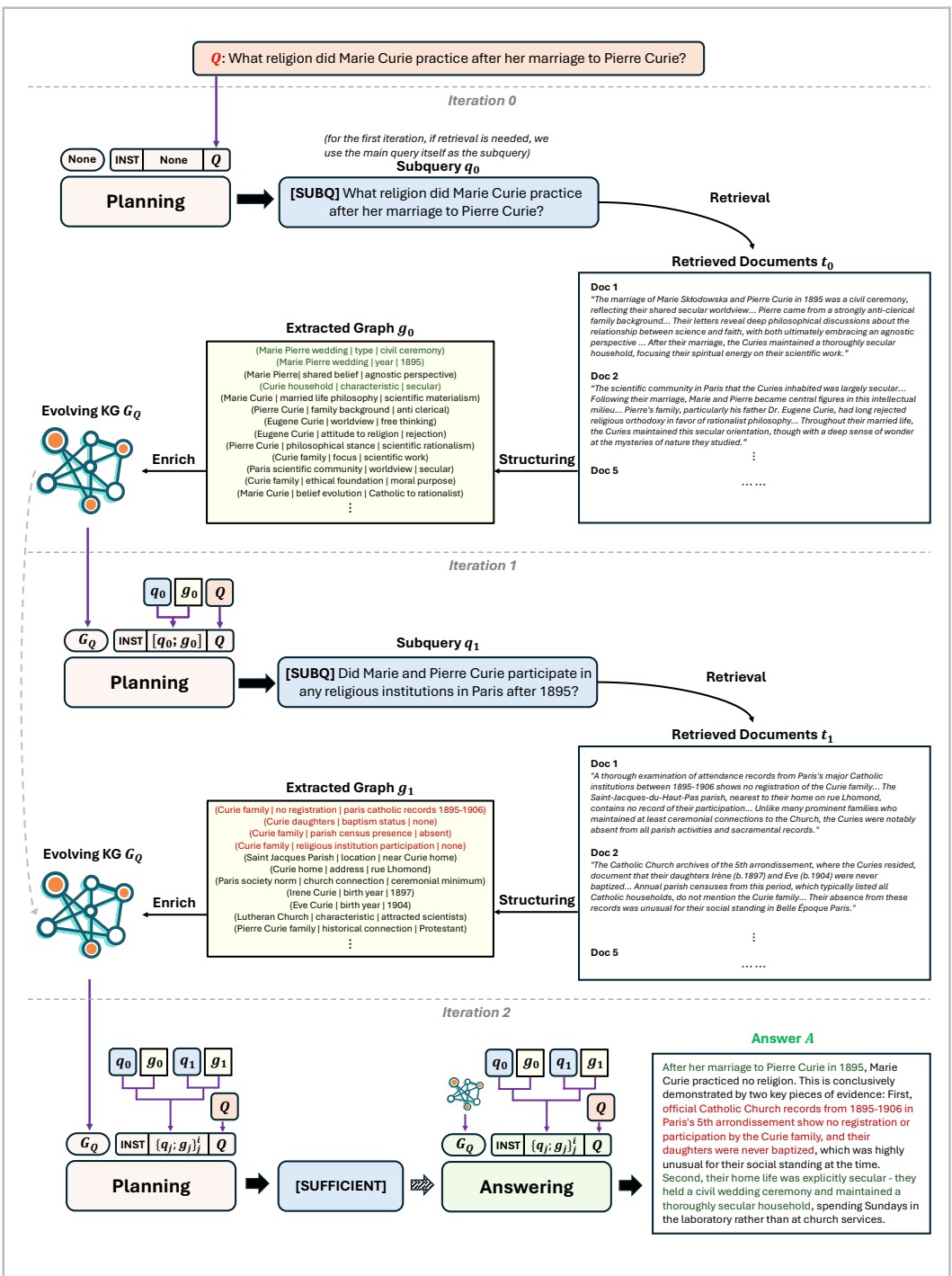

Figure 25: A Running Example of RAS.

## J HYPERPARAMETER STUDY

Table 8: Hyper-parameter study. We **highlight** the setting used in experiments.

| Hyper-parameter | Studied Values |
|---|---|
| **Text-to-Triples Model** | |
| Batch size | {2, 4, **8**, 16, 32} |
| Learning rate | {1e-5, 2e-5, **5e-5**, 1e-4, 2e-4} |
| **GraphLLM (Action Planner & Answerer)** | |
| *GNN Setting (for Graph Token)* | |
| GNN architecture | {GCN, GAT, **Graph Transformer**} |
| Hidden dimension | {512, 768, **1024**, 2048} |
| Number of layers | {2, **3**, 4, 5} |
| Number of heads | {4, 6, **8**, 12} |
| Dropout rate | {0.05, **0.1**, 0.2, 0.3} |
| Projector intermediate dimension | {1024, **2048**, 4096} |
| Projector output dimension | **4096** |
| *LoRA Setting* | |
| LoRA rank (r) | {4, **8**, 16, 32} |
| LoRA alpha | {8, **16**, 32} |
| LoRA dropout | {0.01, **0.05**, 0.1, 0.15} |
| *General Setting* | |
| Learning rate | {1e-6, 2e-6, **5e-5**, 1e-4} |
| Batch size (training) | {**2**, 4, 8, 16, 32} |
| Batch size (inference) | **1** |
| Weight decay | {0.001, **0.01**, 0.05, 0.1} |
| Gradient accumulation steps | {**2**, 4, 8, 16} |
| Gradient clipping | {0.1, 0.3, **0.5**, 1.0} |
| Warmup ratio | {0.05, 0.1, **0.15**, 0.2} |
| Max text length | **2500** |
| Max new tokens | Task-specific (see Appendix F.2) |
| **Others** | |
| Dense retrieval top-$k$ | **5** |

## K   Broader Impacts, Safeguards, and Assets

**Broader Impacts.** The RAS framework presents several promising impacts for both research and practical applications. By transforming unstructured retrieved content into structured, question-specific knowledge graphs, RAS improves reasoning transparency and factual reliability—qualities that are especially valuable in education, scientific research, and technical writing. The ability to explicitly organize retrieved knowledge into interpretable structures also opens up new possibilities for human-in-the-loop AI systems, enabling users to trace and verify model reasoning paths.

However, several ethical considerations remain. RAS's reliance on structured knowledge representations means its effectiveness may be limited in low-resource languages or domains with sparse data. Additionally, biases present in training data can be preserved or even amplified in structured forms. Addressing these concerns will require rigorous evaluation across diverse user groups and careful curation of source corpora. Promoting fairness, transparency, and inclusivity should be core design goals for future RAS-based systems.

**Safeguards.** To mitigate potential harms, we propose several safeguards during the development and deployment of RAS-based systems. First, developers should routinely audit training and retrieval corpora for biases and factual inconsistencies, and apply data augmentation or counterfactual generation techniques where appropriate. Second, regular evaluations should be conducted across a diverse range of tasks, languages, and user demographics to assess any unintended performance disparities. Third, when applied in sensitive domains, human-in-the-loop oversight should be incorporated to validate the structured knowledge outputs and final generations. Lastly, transparency measures, such as providing users with access to intermediate structured knowledge (e.g., the constructed knowledge graph), can help foster trust and enable error analysis.

**Assets Used.** To support the development and evaluation of the RAS framework, we utilized a wide range of publicly available datasets, open-source tools, and pretrained models. All datasets were selected based on relevance to knowledge-intensive tasks, including question answering and long-form generation, and were licensed for academic research or distributed under permissive open-source terms such as MIT, Apache 2.0, and CC BY. Our training and evaluation relied on benchmark datasets such as HotpotQA, ASQA, TriviaQA, and ARC, while structured knowledge was derived using models trained on WikiOfGraph. We used open-source infrastructure including FAISS for retrieval, vLLM for inference optimization, and Graph LLM as our model backbone. For baseline comparisons, we incorporated implementations from Self-RAG, RPG, and ALCE. Closed-source models like Claude-3.5-Sonnet were used for performance comparison and triple extraction under API constraints. Full licensing and usage details are provided in Table 9.

| Asset | Source / URL | License | Usage |
|-------|-------------|---------|-------|
| HotpotQA | https://hotpotqa.github.io/ | MIT | Base QA dataset |
| HotpotQA-SUBQ | We constructed from HotpotQA | MIT (inherited) | Subquery training dataset |
| ASQA | https://github.com/google-research/language/tree/master/language/asqa | Apache 2.0 | Training and evaluation |
| Arc-Easy | https://allenai.org/data/arc | AI2 Research License | Training (subset) |
| TriviaQA | https://nlp.cs.washington.edu/triviaqa/ | CC BY-SA 4.0 | Evaluation |
| PopQA | https://huggingface.co/datasets/akariasai/PopQA | CC BY 4.0 | Evaluation |
| 2WikiMultihopQA | https://github.com/Alab-NII/2wikimultihop | Apache 2.0 | Evaluation |
| PubHealth | https://huggingface.co/datasets/bigbio/pubhealth | MIT | Evaluation |
| ARC-Challenge | https://allenai.org/data/arc | AI2 Research License | Evaluation |
| ELI5 | https://huggingface.co/datasets/eli5 | Apache 2.0 | Evaluation |
| Wikipedia 2018/2020 | https://github.com/facebookresearch/atlas | Creative Commons Public Licenses | Knowledge source for retrieval |
| WikiOfGraph | https://github.com/daehuikim/WikiOFGraph | CC BY 4.0 | Triple model training |
| Claude-3.5-Sonnet | https://www.anthropic.com | Proprietary | Default closed source model |
| LLaMA-2/3 | https://ai.meta.com/llama | Meta LLaMA Community License | Default open source model |
| LLaMA-3.2-3B | Derived from Meta LLaMA | Meta LLaMA Community License | Triple extraction |
| Flan-T5-Large | https://huggingface.co/google/flan-t5-large | Apache 2.0 | Triple extractor |
| Graph LLM | Modified from https://github.com/XiaoxinHe/G-Retriever | Research use | Model backbone |
| Faiss | https://github.com/facebookresearch/faiss | MIT | Dense retrieval |
| vLLM | https://github.com/vllm-project/vllm | Apache 2.0 | Optimized inference |
| Sentence-BERT | https://www.sbert.net | Apache 2.0 | Graph node/edge embedding |
| Self-RAG Code | https://github.com/AkariAsai/self-rag | MIT | Baseline eval |
| RPG Code | https://github.com/haruhi-sudo/RPG | Research use | Baseline eval |
| ALCE Code | https://github.com/princeton-nlp/ALCE | MIT | Eval metrics |

Table 9: Assets used in RAS. All assets are licensed or cited for research use or open-source compatibility.

