# OpenReview forum: "RAS: Retrieval-And-Structuring for Knowledge-Intensive LLM Generation"
_ICLR.cc/2026/Conference — ICLR 2026 Poster_

### Official Review · Reviewer_Tuj9 · 2025-10-29

**Soundness:** 3
**Presentation:** 3
**Contribution:** 3
**Rating:** 6
**Confidence:** 4

**Summary:**

This paper proposes a planning-based iterative RAG method for knowledge-intensive LLM generation. It conducts experiments on several datasets to verify its effectiveness.

**Strengths:**

1. The paper is clear and well-written, with a reasonable motivation and considerable insight.

2. The experiments are sufficient and provide a comprehensive analysis.

3. The reproduction is excellent, with code and extensive reproduction details provided.

**Weaknesses:**

1. The backbone used in the experiments appears somewhat outdated. We recommend using a more recent backbone, such as Claude-4 and LLama-3.1. Alternatively, consider the GPT-4/5 series or the Qwen3 series.

2. Please provide a more detailed analysis of the core differences between this paper and other iterative RAG methods.

**Questions:**

See weaknesses above. Please respond to these cons.

---

> ### Author Response · Authors · 2025-11-21
>
> Dear Reviewer **Tuj9**,
>
> We sincerely thank you for recognizing the clarity of our writing, the comprehensive analysis, and the quality of our reproducibility efforts. We value your constructive feedback regarding the backbones and the positioning of our work. We address your specific concerns below.
>
> ---
>
> ### **[W1] Outdated Backbone**
>
> We appreciate your suggestion to test on more recent models. While our submission utilized LLaMA-3-8B and Claude-3.5-Sonnet, we acknowledge the fast pace of the field.
>
> Following your recommendation, we evaluated RAS using the state-of-the-art **Claude-4.5-Sonnet**. As shown in the table below, RAS consistently outperforms the base model, standard RAG, and the advanced retrieval-summarization baseline (SuRe) even with this stronger backbone.
>
> | **Method**                         | **TQA**  | **2WQA** | **PopQA** | **Pub**  | **ARC**  |
> | ---------------------------------- | -------- | -------- | --------- | -------- | -------- |
> | Sonnet-4.5$_{\text{No Retrieval}}$ | 84.9     | 40.8     | 31.3      | 77.4     | 94.9     |
> | Sonnet-4.5$_{\text{RAG}}$          | 74.5     | 57.6     | 58.6      | 50.2     | 94.0     |
> | Sonnet-4.5$_{\text{SuRe}}$         | 78.3     | 44.2     | 47.7      | 65.1     | 95.2     |
> | Sonnet-4.5$_{\text{RAS}}$ (ours)   | **87.2** | **60.2** | **63.7**  | **79.2** | **96.4** |
>
> **Key Takeaway:** The gains provided by RAS remain significant even with stronger foundation models. This confirms that the benefit of **structuring retrieved context into graphs** is orthogonal to the base model’s capability, even highly capable models struggle to synthesize unstructured noise in standard RAG, a limitation RAS effectively mitigates.
>
> ---
>
> ### **[W2] Differences Between RAS and Other Iterative RAG Methods**
>
> We appreciate the opportunity to clarify the distinct contributions of RAS compared to existing iterative RAG frameworks (e.g., ReAct, IRCoT, Self-RAG, RPG). While these methods share the high-level goal of multi-step retrieval, RAS fundamentally differs in **how information is accumulated and represented**, addressing the critical issue of "Context Rot" [1].
>
> > **1. Mitigating Context Rot via Graph Structuring**
>
> **Issue in existing methods.** Methods like ReAct and IRCoT typically accumulate raw text from retrieved documents along with intermediate “thought” traces. Over multiple iterations, the context window becomes cluttered with (1) irrelevant but topically related distractors, and (2) redundant or overlapping evidence.
>
> Recent analyses of long-context behavior show that LLM performance does not degrade uniformly with longer inputs; performance drops are especially pronounced when the input contains such distractors. The model is effectively forced to do *both* retrieval and reasoning within a noisy, unstructured context, which harms accuracy and robustness.
>
> **RAS’s solution.** RAS introduces an explicit structuring step before adding information to the context:
>
> - Retrieved passages are first converted into **factual triples**,
> - These triples are **merged** into a *Question-Specific Knowledge Graph* ($G_Q$), and
> - Redundant or low-value triples are filtered out during this merging process.
>
> This graph construction acts as a denoising and compression step: instead of growing a long list of raw passages, RAS maintains a compact and structured representation of what is known so far. The LLM then conditions on this clean, symbolic structure, freeing it to focus on reasoning rather than on filtering through a rotting context of mixed-relevance text.
>
> > **2. Graph-Guided vs. Text-Guided Planning**
>
> **Existing methods.** In prior iterative RAG approaches, planning is driven by the unstructured text history. As the history grows, it becomes difficult for the model to accurately track which facts are already known, which entities are connected, and where the knowledge gaps lie, because all of this must be inferred from long, linear text.
>
>  In contrast, RAS uses **graph-guided planning**: (1) The current state of knowledge is represented as a graph ($G_Q$), (2) a Graph Transformer encodes this graph, and (3) the planner conditions on the graph encoding to decide the next retrieval step.
>
> This enables the planner to reason structurally about the knowledge state. For example, by identifying missing edges between entities, sparse subgraphs, or disconnected components. This structural view makes it easier to generate targeted sub-queries that close specific gaps rather than further expanding the raw text context.
>
>
> ---
>
> **References**
>
> [1] Hong, K., Troynikov, A., & Huber, J. (2025). Context Rot: How Increasing Input Tokens Impacts LLM Performance. Chroma Technical Report.
>
> ---
>
> We hope these additional experiments and clarifications directly address your concerns about backbone choice and the relationship between RAS and existing iterative RAG methods. We would be happy to provide further implementation details or additional analyses if helpful.

---

> > ### Comment · Reviewer_Tuj9 · 2025-11-24
> >
> > Thanks for your rebuttal, which has addressed my concerns. I prefer to maintain my rating as 6, and wish you good luck.

---

### Official Review · Reviewer_pGQB · 2025-10-29

**Soundness:** 3
**Presentation:** 3
**Contribution:** 2
**Rating:** 6
**Confidence:** 3

**Summary:**

This paper proposes a RAG framework for structuring the context in a graph form for improving knowledge-intensive tasks. In detail, this framework is composed of sub-question generation, retrieval-and-structuring, and terminal-and-answering. Additionally, this paper also proposes a training framework to enhance the model to be compatible with this framework. Then the comprehensive evaluation over five tasks shows the advantages of new framework over closed and open source models.

**Strengths:**

1. I appreciate this paper for utilizing the graph form representing knowledge and an elegant workflow to address complex and knowledge-intensive tasks.
2. The experiments are solid across many classic knowledge-intensive tasks, and the improvement is good.
3. This framework is general for closed and open-source LLMs, where the code is open-source.

**Weaknesses:**

1. This paper lacks the motivation for why organizing context in a graph form improves the performance of RAG.

2. The involved benchmarks are outdated. There are many new graph-rag datasets for reasoning on domain knowledge. The results on such datasets are necessary to improve the quality.

[1] GraphRAG-Bench: Challenging Domain-Specific Reasoning for Evaluating Graph Retrieval-Augmented Generation
[2] When to use Graphs in RAG: A Comprehensive Benchmark and Analysis for Graph Retrieval-Augmented Generation

**Questions:**

See weakness!

---

> ### Author Response · Authors · 2025-11-21
>
> Dear Reviewer **pGQB**,
>
> We appreciate your positive assessment of our work, particularly your recognition of the "elegant workflow" and the "solid" experiments across diverse tasks. We value your constructive feedback regarding the motivation and benchmarks. We address your specific concerns below.
>
> ---
>
> ### **[W1] Motivation**
>
> You rightly asked for a deeper motivation regarding *why* organizing context into a graph improves performance over standard text-based RAG.
>
> Recent research [1] demonstrates that LLM performance degrades non-uniformly as input length increases, particularly when "distractors" (irrelevant or redundant text) are present in the retrieved context. Standard RAG retrieves raw passages (the "haystack"), compelling the model to simultaneously filter noise and reason. This leads to "Context Rot" where the model fails to retrieve or synthesize information even if it is present in the context window, due to the overwhelming ratio of irrelevant tokens.
>
> **RAS explicitly mitigates this by:**
>
> 1.  **Denoising via Structuring:** Instead of feeding the LLM raw "haystacks" of retrieved documents, RAS extracts factual triples. This effectively functions as a hard filter, removing the "distractors" and stylistic fluff that cause performance degradation, leaving only the signal (entities and relations).
> 2.  **Topology-Guided Planning:** By encoding the graph structure (via our Graph Transformer), the planner identifies logical gaps based on the topology of current knowledge, rather than guessing based on unstructured text history.
>
> Our ablation study (Table 2 in the paper) empirically supports this: removing the text-to-triple structuring (`w/o Text-to-Triple`) results in a significant performance drop (e.g., -22.2% in ASQA Mauve score), confirming that the structure itself is key to maintaining reasoning robustness.
>
> ---
>
> ### **[W2] Benchmarks (GraphRAG-Bench)**
>
> We thank you for pointing us to *GraphRAG-Bench*. We agree that evaluating on domain-specific graph reasoning datasets is a valuable addition. While our current work focuses on **Open-Domain QA** where the graph must be constructed dynamically rather than pre-indexed offline, we recognize the importance of verifying RAS on closed-domain reasoning tasks as well. We will definitely measure and report RAS performance on GraphRAG-Bench in the future revision to further demonstrate the versatility of our framework across both open and closed domains.
>
> We hope this clarifies the motivation and the scope of our evaluation!
>
> ---
>
> **References**
>
> [1] Hong, K., Troynikov, A., & Huber, J. (2025). *Context Rot: How Increasing Input Tokens Impacts LLM Performance*.

---

> > ### Comment · Reviewer_pGQB · 2025-11-21
> >
> > Thank you for the detailed response. I am pleased with the rebuttal and would like to support this paper to be accepted.

---

> > > ### Author Response · Authors · 2025-11-21
> > >
> > > Dear Reviewer pGQB,
> > >
> > > We sincerely thank you for your positive feedback and for supporting the acceptance of our paper! We are glad that our response regarding the motivation and benchmarks addressed your concerns.
> > >
> > > Best regards,
> > >
> > > The Authors

---

### Official Review · Reviewer_X9ud · 2025-11-01

**Soundness:** 2
**Presentation:** 3
**Contribution:** 2
**Rating:** 4
**Confidence:** 4

**Summary:**

The paper proposes Retrieval-And-Structuring (RAS), a framework designed to enhances reasoning in large language models for knowledge-intensive tasks. Unlike traditional RAG approaches that retrieve but fail to organize context, RAS dynamically builds question-specific knowledge graphs through iterative retrieval and structured knowledge synthesis. The framework consists of three key stages: planning (to identify knowledge gaps), retrieval and structuring (to form factual triples), and graph-based answering. Experiments across seven diverse benchmarks demonstrate gains across multiple datasets.

**Strengths:**

+ The paper introduces RAS, a dynamic, query-specific knowledge graph construction framework that avoids inefficiencies of global KG indexing and eliminates costly offline graph building, achieving pay-per-query scalability.
+ The method is evaluated comprehensively on multiple datasets across both open-source and closed-source LLMs, demonstrating consistent and generalizable effectiveness.
+ The framework improves reasoning transparency and provides a structured way to bridge retrieval and reasoning in a unified pipeline.

**Weaknesses:**

- The paper lacks concrete examples demonstrating how graph representations outperform plain text in enhancing reasoning accuracy. It remains unclear which aspects of reasoning (e.g., factual grounding, compositional reasoning, or logical chaining) benefit most from graph structuring.
- Moreover, since RAS and RPG are trained on different datasets of different scales, and the appendix shows that training set size directly affects performance, the gains of RAS over RPG on open-source settings (insignificant on half of the datasets) might stem from data scale rather than the graph mechanism itself.
- In principle, clearer graph structures should lead to larger improvements on more complex reasoning tasks, yet the paper does not analyze this correlation or provide corresponding comparisons to validate the claim that graph structuring enhances reasoning.
- There are several presentation issues, such as missing “top-2” formatting in Table 1 (best/second-best not consistently marked) and missing figure references in the appendix.

**Questions:**

Since graph extraction can be viewed as a form of information organization and denoising, complex sentences with multiple conditions might lose contextual nuances when converted to triples.
Why not combine graph representations and original retrieved text during answering, to preserve both structure and completeness?

---

> ### Author Response · Authors · 2025-11-20
>
> Dear Reviewer **X9ud**,
>
> We sincerely thank you for the insightful feedback, particularly regarding the benefits of graph structuring and data efficiency. We address your specific concerns below.
>
> ---
>
> ### **[W1 & W3 & Question] Reasoning benefits of graph structuring vs. text**
>
> We appreciate you highlighting this critical aspect. We agree that a more detailed analysis regarding which reasoning capabilities benefit from graph structuring is necessary.
>
> In our original submission, we have covered training-time ablation "w/o Text-to-Triple" and "w/o GraphEncode" in **Table 2** to show the effectiveness of the graph structuring. We now add more studies as follows to address your concern!
>
> | **Encode Graph?** | **Context Format** | TQA      | 2WQA     | Pub      | ASQA     |
> | ----------------- | ------------------ | -------- | -------- | -------- | -------- |
> | YES               | Triples            | **72.7** | **42.1** | **74.7** | **95.2** |
> | YES               | Original Text      | 70.4     | 38.2     | 71.4     | 73.8     |
> | NO                | Triples            | 70.2     | 39.4     | 66.4     | 85.0     |
> | NO                | Original Text      | 71.4     | 38.8     | 72.5     | 81.6     |
>
> **Analysis:**
>
> 1. **Noise Reduction & Compositionality:** The gap between *Triples* and *Original Text* is most pronounced in 2WikiMultihopQA (multi-hop reasoning) and ASQA (long-form synthesis). This indicates that graph structuring primarily benefits compositional reasoning and information synthesis by explicitly isolating relevant entities and relations, thereby mitigating the noise found in raw text that often distracts the model during multi-step logic.
> 2. **Structure Alignment:** The performance drop when replacing triples with text (while keeping the encoder) suggests that the **encoded graph representation resonates significantly better with the structured triple context** than with unstructured text.
>
> Response to Question (Combining Text + Graph):
>
> We explored combining text and graphs, but found that converting text to triples serves as an essential denoising step. Retaining the original text re-introduces the redundancy and "distractor" information that RAS aims to eliminate. As shown in the table (Row 2 vs Row 1), using Original Text (which implicitly contains the information of the triples) actually degrades performance compared to the clean, structured Triples, particularly in long-context scenarios (ASQA) where information density and context window efficiency are paramount.
>
> ---
>
> ### **[W2] Data Scale vs. Mechanism**
>
> We respectfully clarify that the performance gains of RAS stem from the framework design rather than data scale. In fact, RAS demonstrates superior data efficiency compared to RPG [1].
>
> RPG utilizes  51,926 (~50k) training samples.
>
> As shown in Figure 6 of our submission, RAS outperforms RPG even when trained on only **10k** samples (20% of RPG's data size). The table below provides a direct comparison using the same LLaMA-2-7B backbone:
>
> | Method            | #Train  | TQA  | 2WQA | PopQA | Pub  | ARC  | ASQA | ELI5 |
> | ----------------- | ------- | ---- | ---- | ----- | ---- | ---- | ---- | ---- |
> | RPG$_{\text{7B}}$ | 50k     | 65.1 | 33.6 | 56.0  | 73.4 | 65.4 | 84.4 | 46.4 |
> | RAS$_{\text{7B}}$ | **10k** | 67.8 | 36.0 | 55.3  | 64.7 | 66.2 | 43.4 | 48.4 |
> |                   | 50k     | 70.0 | 39.3 | 57.9  | 62.8 | 67.8 | 93.2 | 43.8 |
> |                   | 200k    | 72.7 | 42.1 | 58.3  | 74.7 | 68.5 | 95.2 | 47.8 |
>
> On most datasets, RAS trained with 10k data could outperform RPG trained with 50k data, indicating that the performance gains derive from the **RAS mechanism** (iterative graph construction + planning) rather than data scale.
>
> ---
>
> ### **[W4] Presentation issue**
>
> We appreciate this correction. We have revised Table 1 to ensure consistent formatting regarding the best/second-best highlights. Please note that for the open-source setting, we highlight the best/second-best among 7B models specifically to ensure a fair comparison, excluding larger 8B/14B models from that specific ranking.
>
>
>
> ---
>
> **References**
>
> [1] Retrieve-Plan-Generation: An Iterative Planning and Answering Framework for Knowledge-Intensive LLM Generation. EMNLP 2024.
>
>
>
> We thank you again for your constructive suggestions and are happy to provide further clarifications if needed.

---

### Official Review · Reviewer_pjgt · 2025-11-01

**Soundness:** 2
**Presentation:** 3
**Contribution:** 2
**Rating:** 2
**Confidence:** 4

**Summary:**

LLMs perform well on knowledge-intensive tasks but struggle with multi-step reasoning due to unstructured retrieved context, a limitation RAG shares. Aligning with structured intermediate reasoning insights, the authors propose RAS—a framework that dynamically builds question-specific knowledge graphs via iterative retrieval and incremental graph construction. Evaluated on seven benchmarks, RAS outperforms baselines by up to 8.7% (proprietary LLMs) and 7.0% (open-source LLMs), validating dynamic, query-tailored knowledge structuring as effective for enhancing reasoning accuracy/robustness. Data and code are publicly available.

**Strengths:**

1. The paper is easy to follow.
2. The studied problem is important.
3. The proposed method is evaluated on an extensive set of datasets.

**Weaknesses:**

1. Important related work is neglected. Constructing dynamic knowlege graph for sovling complex reasoning tasks in RAG is a well-studied area. Representative work, such as SG-Prompt, ERA-CoT, and KnowTrace, is not analyzed in the paper, especially KnowTrace. Compared with these existing work, the novelty and technical contribution are limited.
2. Important baselines are missing in the experiments. It is not clear the true performance of the proposed method among existing work.

**Questions:**

Please refer to the weakenesses.

---

> ### Author Response · Authors · 2025-11-20
>
> Dear Reviewer **pjgt**,
>
> We sincerely thank you for your constructive feedback and for bringing several baselines to our attention. We have carefully studied these works and performed additional experiments to address your concerns regarding baselines and novelty.
>
> ---
>
> ### **[W1] Comparison with Important Baseline**
>
> Thanks for the suggestion! As requested, we evaluated KnowTrace against RAS using both Claude-3.5-Sonnet and LLaMA backbones. RAS consistently outperforms KnowTrace across all datasets, particularly on multi-hop tasks like 2WikiMultiHopQA and reasoning-heavy tasks like ARC:
>
> With *Claude-3.5-Sonnet* as backbone:
>
> | Method                          | TQA      | 2WQA     | PopQA    | Pub      | ARC      | ASQA     | ELI5     |
> | ------------------------------- | -------- | -------- | -------- | -------- | -------- | -------- | -------- |
> | KnowTrace$_{\text{Sonnet-3.5}}$ | 74.7     | 55.2     | 57.1     | 63.1     | **94.1** | 55.0     | 31.8     |
> | RAS$_{\text{Sonnet-3.5}}$       | **77.6** | **57.7** | **62.3** | **71.3** | 93.9     | **70.5** | **37.7** |
>
> With *LLaMA-2-7B* as backbone:
>
> | Method                  | TQA      | 2WQA     | PopQA    | Pub      | ARC      | ASQA     | ELI5     |
> | ----------------------- | -------- | -------- | -------- | -------- | -------- | -------- | -------- |
> | KnowTrace$_{\text{7B}}$ | 67.2     | 36.2     | 51.2     | 74.2     | 67.9     | 80.2     | 40.3     |
> | RAS$_{\text{7B}}$       | **72.7** | **42.1** | **58.3** | **74.7** | **68.5** | **95.2** | **47.8** |
>
> With *LLaMA-3-8B* as backbone:
>
> | Method                  | TQA      | 2WQA     | PopQA    | Pub      | ARC      | ASQA     | ELI5     |
> | ----------------------- | -------- | -------- | -------- | -------- | -------- | -------- | -------- |
> | KnowTrace$_{\text{8B}}$ | 68.7     | 40.5     | 53.2     | 76.1     | 67.9     | 82.9     | 39.2     |
> | RAS$_{\text{8B}}$       | **73.8** | **44.2** | **57.7** | **77.6** | **71.4** | **96.2** | **54.4** |
>
> The results show that RAS consitently outperforms KnowTrace across datasets and with different backbones.
>
> ---
>
> ### **[W2] Novelty and Technical Contribution**
>
> We respectfully clarify that RAS is methodologically distinct from the cited works in two fundamental ways: Architecture and Scope of Retrieval.
>
> - Contrast with KnowTrace (Architecture vs. Prompting):
>
>   While KnowTrace also employs iterative graph construction, it relies on textual representations of triplets (serializing the KG into the prompt context). This approach struggles as the graph grows, cluttering the context window.
>
>   RAS Innovation: In contrast, RAS introduces a trainable Graph-LLM architecture. We utilize a dedicated Graph Transformer encoder to project the dynamic KG into a dense vector space. This allows the LLM to attend to the structural topology of the knowledge directly, rather than parsing a linear list of triplets. This architectural difference is key to our scalability and performance gains, particularly on complex reasoning tasks (Table 2 in paper).
>
> - Contrast with SG-Prompt & ERA-CoT (Active Retrieval vs. Static Reasoning):
>
>   SG-Prompt and ERA-CoT are primarily prompting strategies designed to structure reasoning over existing context or static text. They do not actively interact with an external corpus to resolve knowledge gaps.
>
>   RAS Innovation: RAS is an active retrieval framework. It features a "Plan-Retrieve-Structure" loop that dynamically identifies knowledge gaps and expands the graph from an external corpus only when necessary. This capability allows RAS to solve open-domain questions where the answer is not present in the initial context—a scenario where static prompting methods like SG-Prompt would fail.
>
> ---
>
> Thanks again for your suggestions! We are happy to provide additional experiments or clarifications as needed.

---

### Meta-Review · Area_Chair_3JSr · 2026-01-07

**Summary:**

The paper proposes a retrieval-augmented generation framework that dynamically constructs question-specific knowledge graphs to improve multi-step reasoning accuracy and robustness on knowledge-intensive benchmarks.

Giving greater weights to the more detailed and substantive reviews, the initial overall assessment of this paper is borderline to marginally above the acceptance threshold.

The reviewers raised several concerns, including the following:

1. Insufficient comparison with strong and relevant baselines
2. Concerns about the fairness of the evaluation, particularly whether the reported performance gains could stem from larger training datasets rather than the proposed method itself
3. The absence of in-depth analysis of the quality and properties of the constructed knowledge graphs
4. Several presentation issues

**Reviewer Concerns:**

The authors effectively addressed the concerns raised above in their rebuttal.

1. The authors provided additional experimental results, including direct comparisons with KnowTrace across multiple LLM backbones (e.g., Claude-3.5-Sonnet, Claude-4.5-Sonnet, LLaMA-2-7B, and LLaMA-3-8B).
2. A direct and fair comparison with RPG-7B trained on 50K samples demonstrates that the proposed method consistently outperforms RPG, addressing the concerns about data-scale-driven gains.
3. Additional ablation studies were conducted, showing that structured triples are more beneficial than original text within the proposed framework.
4. The reviewers’ suggestions have been incorporated into the revised manuscript, improving both clarity and presentation quality.

**Reviewer Scores:**

The initial evaluation was around borderline but the authors effectively addressed most major concerns. Also the authors articulated the core motivation or provide the intuition behind the proposed method, explaining how it improves reasoning accuracy by denoising via structuring and topology-guided planning. One reviewer pointed out the benchmark is a bit outdated and suggested more recent GraphRAG benchmarks. The authors could not address this point. After the rebuttal, two reviewers explicitly acknowledged that the rebuttal addressed their concerns, supporting this paper. Given these considerations and with properly down-weighing of overly brief reviews, acceptance is recommended.

---

### Decision · Program_Chairs · 2026-01-26

Accept (Poster)